

# Atmospheric aerosol compositions over the South China Sea: Temporal variability and source apportionment

Hong-Wei Xiao[1,2], Hua-Yun Xiao[1,2*], Li Luo[1,2*], Chun-Yan Shen[3], Ai-Min Long[4], Lin Chen[5], Zhen-Hua Long[5],
Da-Ning Li[5]

[1]Laboratory of Atmospheric Environment, Key Laboratory of Nuclear Resources and Environment (Ministry of Education), East China University of Technology, Nanchang 330013, China

[2]School of Water Resources and Environmental Engineering, East China University of Technology, Nanchang 330013, China

[3]College of Fisheries, Guangdong Ocean University, Zhanjiang 524088, China

[4]State Key Laboratory of Tropical Oceanography, South China Sea Institute of Oceanology, Chinese Academy of Sciences, Guangzhou 510301, China

[5]Xisha Deep Sea Marine Environment Observation and Research Station, South China Sea Institute of Oceanology, Chinese Academy of Sciences, Sansha 573199, China

*Correspondence to: Hua-Yun Xiao (xiaohuayun@ecit.cn) and Li Luo (luoli@ecit.cn)

**Abstract.** Major inorganic chemical ionic concentrations ($Na^+$, $Cl^-$, $SO_4^{2-}$, $Ca^{2+}$, $Mg^{2+}$, $K^+$, $NH_4^+$, $NO_3^-$) were determined in total suspended particulates (TSP) at Yongxing Island in the South China Sea (SCS), from March 2014 to February 2015. The annual average concentration of TSP was $89.6 \pm 68.0$ μg/m³, with $114.7 \pm 82.1$, $60.4 \pm 27.0$, and $59.5 \pm 25.6$ μg/m³ in cool, warm, and transition seasons, respectively. $Cl^-$ had the highest concentration, with an annual average of $7.73 \pm 5.99$ μg/m³, followed by $SO_4^{2-}$ ($5.54 \pm 3.65$ μg/m³), $Na^+$ ($4.00 \pm 1.88$ μg/m³), $Ca^{2+}$ ($2.15 \pm 1.54$ μg/m³), $NO_3^-$ ($1.95 \pm 1.34$ μg/m³), $Mg^{2+}$ ($0.44 \pm 0.33$ μg/m³), $K^+$ ($0.33 \pm 0.22$ μg/m³), and $NH_4^+$ ($0.07 \pm 0.07$ μg/m³). Concentrations of TSP and the major ions showed seasonal variations, higher in the cool season and lower in the warm and transition seasons, which were influenced by wind speed, temperature, relatively humidity, rain, and air masses. Back trajectories, concentration weighted trajectories (CWT), and positive matrix factorization (PMF) of chemical compositions were analyzed for source apportionment, source contribution, and spatio-temporal variation of major ions. Back trajectories and CWTs showed that air masses at Yongxing Island were mainly from



the northeast, southwest, and southeast in the cool, warm, and transition seasons, respectively. $Na^+$ and $Cl^-$ were mainly from sea salt, which made up 74% and 82%, respectively. Asian dust contributed 50% of $Ca^{2+}$ to the marine aerosols. Anthropogenic sources were very important for atmospheric aerosols over the island. Fossil fuel combustion (especially coal in Chinese coastal regions) was the important sources of $NO_3^-$ (56%) and $SO_4^{2-}$ (22%). Biomass burning in Asia accounted for 41% of $K^+$. 69% of $NH_4^+$ and 38% of $SO_4^{2-}$ were of marine biogenic sources.

**Keywords:** Source apportionment; dust; biomass burning; fossil fuel combustion; marine biogenic source

## 1 Introduction

Aerosols or particulate matter potentially affect global atmospheric processes, chemistry, cloud formation, acid and nutrient deposition in sensitive ecosystems, and affect human health as air pollution (Deng et al., 2010;
Zhang et al., 2015). Aerosols have complex sources, including primary aerosols direct from natural and anthropogenic sources, such as terrestrial dust from weathering, sea spray, biomass burning, and biological emissions. There are also secondary aerosols, which form from condensable atmospheric gases such as $SO_2$, $NO_x$, and $NH_3$ (Kolb and Worsnop, 2013; Xiao et al., 2012a, 2014). Therefore, aerosols are composed of various complex chemical components (Xiao and Liu, 2004) and contain sulfate, nitrate, ammonium, and mineral elements (Zhang
et al., 2007). Because of rapid economic and industrial development in the last few decades, many aerosols and their precursors released by human activities have become a major environmental problem worldwide (Kolb and Worsnop, 2013). Aerosols from anthropogenic emissions can be transported long distances from polluted regions to remote open oceans, which is well recognized as a major pathway for the supply of anthropogenic material to ocean surface waters (Duce et al., 2008; Lawrence and Lelieveld, 2010; Jung et al., 2012).

Atmospheric aerosol deposition over remote open oceans can increase ocean productivity and carbon sequestration (Duce et al., 2008; Kim et al., 2014; Landing and Paytan, 2010), but aerosol input to such oceans varies considerably in both time and space. Such variations are modulated by sources, chemical processes, and environmental parameters, e.g., wind and temperature (Landing and Paytan, 2010; Xiao et al., 2015). Time-series measurements of aerosol particles over the subarctic Northeast Pacific Ocean have indicated that major
contributions to the aerosol mass were from the oxidation of dimethyl sulfide (DMS), sea salt, and ship emissions (Phinney et al., 2006). High $NH_4^+$ and $NO_3^-$ concentrations have been found in North and South Pacific Ocean when air masses were derived from the Asian continent and the Kamchatka Peninsula, whereas low concentrations were observed when air masses were from the central Pacific Ocean (Jung et al., 2012); therefore, $NH_3$ and $NO_x$ are influenced by both natural and anthropogenic sources. In the North Sea, the anthropogenic fraction is lower for



air masses from the North Atlantic than for those from the European continent (Ebert et al., 2000). There are four discrete transport regimes over the western North Atlantic Ocean at Bermuda that affect the sources (Keene et al., 2005). The East China Sea was more influenced by Asian dust storms whereas the Yellow Sea was more influenced by human activities, during March and April, 2011 (Zhao et al., 2015). Dust from the Gobi Desert with pollutants from eastern China transports numerous dust elements and secondary pollutants to the South China Sea (SCS) (Liu

et al., 2014).

The SCS is in the tropical–subtropical rim of the Northwest Pacific Ocean and is one of the largest marginal seas in the world (Fig. 1). The SCS is adjacent to several rapidly growing Asian economies, including China, the Philippines, Malaysia, Vietnam, and Indonesia (Kim et al., 2014). Consequently, the sea receives substantial amounts of aerosols from surrounding regions through long-range atmospheric transport (Fig. 1; Atwood et al.,

2013; Jung et al., 2012). The SCS has a monsoon climate, with a northeast monsoon in winter and spring and a southwest monsoon in summer and autumn (Cui et al., 2016). Thus, aerosol optical thickness (AOT) shows spatial and seasonal variations, with higher AOT in the northern SCS during the cool season and higher AOT in the southern SCS in the warm season (Fig. 1). The SCS receives dust and pollutions from the northeast SCS in winter and spring, e.g., from China and Japan (Wang et al., 2013; Xiao et al., 2015); during summer and autumn, the SCS

receives aerosols and pollutants from biomass burning in the southwest of SCS, e.g., from Malaysia and Indonesia (Atwood et al., 2013). According to Lawrence and Lelieveld (2010), emissions of $SO_2$ and $NO_x$ in Northeast Asia are mainly from fossil fuel combustion and industrial processes, and from fossil fuel combustion and biofuel burning in South Asia. Biomass burning in Asian countries is an important contributor to aerosol deposition in the SCS (Streets et al., 2003).

To get better understanding of potential sources, source contributions, and spatio-temporal variations of marine aerosols over the SCS, total suspended particulate (TSP) were continuously collected at Yongxing Island from March 2014 to February 2015. The concentrations of major inorganic water-soluble ions were determined. Furthermore, back trajectories, concentration-weighted trajectory (CWT) and positive matrix factorization (PMF) analyses were also used to identify and apportion the main sources of aerosols and their chemical composition over

the SCS.

## 2 Materials and Methods

### 2.1 Study site



Aerosol samples were collected from March 2014 through February 2015 on the rooftop of Xisha Deep Sea

Marine Environment Observation and Research Station, South China Sea Institute of Oceanology, Chinese

Academy of Sciences (SCSIO, CAS). This station is at the Yongxing Island (YXI, Fig. 1; 16.83°N, 112.33°E).

This island has a tropical monsoon climate, with northeast monsoon in winter and spring, and southwest monsoon

in summer and autumn (Xiao et al., 2015). The island is located in the high aerosol concentrations area of the

northern SCS during winter and spring, and also at the periphery of such concentrations area of the southern SCS

during summer and autumn (Fig. 1). Therefore pollutants can be from Northeast Asia in winter and spring, and

Southeast Asia in summer and autumn. It is also influenced by local SCS marine sources. More information about

Yongxing Island is given elsewhere (Xiao et al., 2015 and 2016). In the present study, it was divided into three

seasons according to wind direction and back trajectories. The cool season was March, April, October, November,

December 2014 and January, February 2015, with principal air masses from the northeast, and the warm season

was June–September 2014, with air masses major from the southwest. The transition season was May 2014, with

major air masses from the local southeast. Yongxing Island has an annual average temperature of 27.7 ± 2.7 °C,

relative humidity (RH) of 80 ± 7%, and wind speeds of 3.6 ± 1.8 m/s. There were strong seasonal variations between

March 2014 and February 2015 (Fig. 2). Annual rainfall was 1526 mm during this period, with about 30% occurring

in the cool season (Fig. 2).


## 2.2 Sample collection and chemical analyses

Aerosol was collected on quartz filters (8 × 10 inch, Tissuquartz™ Filters, 2500 QAT-UP, Pallflex,

Washington, USA) using a special high-flow rate (1.05 ± 0.03 m³/min) KC-1000 sampler (Laoshan Institute for

Electronic Equipment, Qingdao, China). The sampling time was nominally 96 hours (4 days one sample). All

samples were stored in a refrigerator at −20°C until analysis in the laboratory.

In the laboratory, one eighth filters were cut and placed in a clean 50-ml Nalgene tube with additional 35-ml

ultrapure water. These tubes were washed for 30 minutes using ultrasonic vibration. They were then shaken for 30

minutes on a horizontal shaker at a rate of ~300 rpm and left to rest for an another 30 minutes at room temperature.

The extract was filtered using pinhole filters, which were then rinsed twice with 5-ml ultrapure water. The extract

and rinse were put into 50-ml tubes together and stored in a refrigerator at −20 °C until chemical analyses.

Major anion concentrations ($F^-$, $Cl^-$, $NO_3^-$, $SO_4^{2-}$, $Br^-$) were determined by ICS-90 ion chromatography (Dionex,

California, USA). Water-soluble metal and nonmetal elemental concentrations (Al, Ca, Fe, K, Mg, Mn, Na, $SiO_2$,

Sr) were analyzed by an MPX inductively coupled plasma optical emission spectrometer (ICP-OES, Vista, CA,



USA). $NH_4^+$ concentration was determined by spectrophotometry after treatment with Nessler's reagent. The

detection limits of $F^-$, $Cl^-$, $NO_3^-$, $SO_4^{2-}$, $Br^-$ were 0.03, 0.03, 0.08, 0.075 and 0.1 mg/L, respectively, and the relatively

standard deviation of these ions were 0.57%, 2.55%, 1.16%, 1.36% and 11.36%, respectively (Xiao et al., 2013

and 2016). The detection limits of Al, Ca, Fe, K, Mg, Mn, Na, $SiO_2$, Sr were 0.025, 0.003, 0.002, 0.06, 0.0005,

0.0005, 0.02, 0.015 and 0.00008 mg/L, respectively, and the relatively standard deviation of these ions were less

than 1.5% (Xiao et al., 2013 and 2016). The detection limit of $NH_4^+$ was 0.1 mg/L and its relatively standard

deviation was less than 5.0% (Xiao et al., 2013 and 2016). In this study, Al in most of samples was less than the

detection limit.

## 2.3 Back trajectories and CWT analysis

Back trajectories and CWT are used to determine the long-distance transport of atmospheric pollutants and

regional source areas. Detailed principles of back trajectories and CWT were given in our previous studies (Xiao

et al., 2014; Xiao et al., 2015). For each day, 10-day (240 hours) back trajectories of air masses arriving at Yongxing

Island were computed. CWT modeled TSP, and $Ca^{2+}$, $Mg^{2+}$, $K^+$, $SO_4^{2-}$, $NO_3^-$ and $NH_4^+$ concentrations at the island.

The region from 70°E to 160°E and from 20°S to 60°N was defined as the source domain based on back trajectories

during the sampling period, containing 14,400 grid cells of $0.5° \times 0.5°$.


## 2.4 PMF model

PMF is an effective source apportionment receptor model that does not require source profiles prior to analysis

and has no limitation on source numbers (Tiwari et al., 2013; Zhang et al., 2015). The principles of PMF are detailed

elsewhere (Han et al., 2006; Hien et al., 2004; Schmale et al., 2013; Yu et al., 2013). In our study, PMF 5.0 (United

States Environmental Protection Agency) was used to determine source apportionment of TSP and each major ion

based on TSP, $F^-$, $Cl^-$, $NO_3^-$, $SO_4^{2-}$, Ca, K, Mg, Mn, Na, and Sr. Five physically realistic sources for TSP and major

ions were identified, i.e., sea salt, crust, biomass burning, fossil fuel combustion, and marine biogenic.

## 3 Results and Discussion

### 3.1 Aerosol chemical composition over SCS and comparison with global marine aerosols

3.1.1 Aerosol characteristics over SCS

Figures 3 and 4 provide information on atmospheric concentrations of TSP aerosols and major inorganic ions

during the sampling period at Yongxing Island. The annual average TSP concentration at the island was 89.6 ±





68.0 µg/m³, with a range of 16.4 to 440.1 µg/m³. This TSP level is much lower than cities around the world (Wang

et al., 2006; Xiao and Liu, 2004; Deng et al., 2011; Naga et al., 2014). It is also lower than some remote sites, such

as Mountain Tai (Deng et al., 2011). Of course, it is higher than lots of remote sites, such as Tianchi, Qinghai Lake

(Deng et al., 2011; Zhang et al., 2014).

        The major inorganic ionic concentrations ($Na^+$, $Cl^-$, $SO_4^{2-}$, $Ca^{2+}$, $Mg^{2+}$, $K^+$, $NH_4^+$, and $NO_3^-$) accounted for 24.8%

of TSP. $Cl^-$ had the highest concentration among these ions, from 0.39 to 36.47 µg/m³, with an annual average of

$7.73 \pm 5.99$ µg/m³. It was followed by $SO_4^{2-}$ (range 0.52–23.34 µg/m³, average $5.54 \pm 3.65$ µg/m³), $Na^+$ (0.9 –8.86

µg/m³, average $4.00 \pm 1.88$ µg/m³), $Ca^{2+}$ (0.17–9.65 µg/m³, average $2.15 \pm 1.54$ µg/m³), $NO_3^-$ (0.10–10.05 µg/m³,

average $1.95 \pm 1.34$ µg/m³), $Mg^{2+}$ (0.02–1.55 µg/m³, average $0.44 \pm 0.33$ µg/m³), $K^+$ (0.06–1.13 µg/m³, average

$0.33 \pm 0.22$ µg/m³), $NH_4^+$ (0.01–0.32 µg/m³, average $0.07 \pm 0.07$ µg/m³) (Fig. 3).

3.1.2 Aerosols over SCS compared with global marine aerosols

        The annual average TSP concentration at Yongxing Island is also lower than those in the northern Yellow Sea

(123.2 µg/m³), another Chinese marginal sea (Wang et al., 2013). However, the mean TSP concentration at

Yongxing Island is not lower than other remote islands or seas, such as the Indian and Pacific oceans and three

islands of Okinawa (Arakaki et al., 2014; Balasubramanian et al., 2013; Zhang et al., 2010; Zhang et al., 2014).

The aforementioned annual average concentrations and orders of ionic concentration are comparable with

those reported in many remote oceans (Fig. 5), e.g., Hedo, which is at the junction of the East China Sea and

Northwest Pacific. The marine ions ($Na^+$ and $Cl^-$) accounted for 53% of total major ions (Fig. 4). $Na^+$ and $Cl^-$

concentrations at Yongxing Island were higher than most reported values in global ocean and remote sites (Fig. 5),

such as the Indian Ocean, Arabian Sea, Oki and Rishiri islands in the Sea of Japan, Amsterdam Island in the

Southern Ocean, Bermuda in the Atlantic Ocean, and Hawaii in the Pacific. However, $Na^+$ and $Cl^-$ concentrations

were lower than samples from cruises, such as over the southern Atlantic and Pacific. The $Ca^{2+}$ concentration was

the highest in all global oceans and composed 10% of major ions, followed by the Mediterranean Sea, southern

Atlantic and northern Atlantic-2 (Fig. 5). The relatively high $Ca^{2+}$ concentration may be because of Asian terrestrial

dust transported to Yongxing Island and dust from local weathered dead coral from Yongxing Island development

(Xiao et al., 2016). As a tracer for dust, non-sea salt $Ca^{2+}$ (nss-$Ca^{2+}$) accounted for 93% of total $Ca^{2+}$, ranging from

0.14 to 9.31 µg/m³ with an annual average of 1.99 µg/m³. Large contributions of nss-$Ca^{2+}$ were also found in the

Mediterranean Sea, southern Atlantic and northern Atlantic-2, being at 88.4%, 90.3% and 90.0%, respectively

(Zhang et al., 2010). The relatively high nss-$Ca^{2+}$ concentrations were potentially from the crust or dust from the



Sahara Desert (Zhang et al., 2010). Comparing Yongxing Island with global oceans, average $Mg^{2+}$ concentrations were nearly consistent with those of $Na^+$ (Fig. 5). $K^+$ was also the highest among the global oceans (Fig. 5). As a tracer for biomass/biofuel burning, nss-$K^+$ ranged from 0 to 0.87 μg/m$^3$, with an annual average of 0.18 μg/m$^3$ and a contribution of 55% to total water soluble $K^+$ at Yongxing. In general, $SO_4^{2-}$, $NO_3^-$ and $NH_4^+$ were major in the form of secondary inorganic aerosols. They accounted for only 34.0% of total inorganic ionic concentrations, giving them an intermediate position among the global ocean (Fig. 5). The average $SO_4^{2-}$ concentration at Yongxing was

the highest among those in the global ocean. As shown in Fig. 4, the mean contribution of $SO_4^{2-}$ to major inorganic ionic components was ~ 25% at Yongxing. The nss-$SO_4^{2-}$ concentration was 3.66 μg/m$^3$, with a contribution of 66.1% to total $SO_4^{2-}$. Similar to $SO_4^{2-}$, the average concentration of $NO_3^-$ in this study was the highest among those in the global ocean. It accounted for 9% of major ions at Yongxing Island. This indicates that a large number of anthropogenic sources affected the concentrations of $SO_4^{2-}$ and $NO_3^-$. It was surprising that $NH_4^+$ had relatively low

concentrations over most oceans, except for the southern Atlantic and Mediterranean Sea (Fig. 5). The average $NH_4^+$ concentration was $0.07 \pm 0.07$ μg/m$^3$ in aerosol at Yongxing Island (Fig. 3), representing < 1% of total major ions (Fig. 4). Further, low concentrations of $NH_4^+$ were also observed in rainwater on the island (Xiao et al., 2016).

### 3.1.3 Global marine aerosol chemical patterns

Globally, sea salt ions ($Na^+$ and $Cl^-$) were the most important components in marine atmospheric aerosol, with higher concentration of $Cl^-$ than $Na^+$, except over the Mediterranean and North seas (Fig. 5). In the marine atmosphere, sea salt aerosol (NaCl) can react with sulfuric acid and nitric acid to release HCl, which results in a deficit of $Cl^-$ relative to $Na^+$ (Zhang et al., 2010). It is also found that a deficit of $Cl^-$ in transition season at Yongxing Island (Fig. 4). For $SO_4^{2-}$ with ss-$SO_4^{2-}$ and nss-$SO_4^{2-}$, nss-$SO_4^{2-}$ was greatly influenced by anthropogenic sources

from developed industrial areas, leading to higher concentrations of $SO_4^{2-}$ than $Na^+$ and $Cl^-$. Examples were Bermuda, Ogasawara, and the Arabian Sea (Fig. 5), where nss-$SO_4^{2-}$ was the preferred species for acid displacement (Zhang et al., 2010). As another import ion of anthropogenic sources, $NO_3^-$ concentrations were often consistent with those of nss-$SO_4^{2-}$ (Zhang et al., 2010), with relatively high concentrations among major ions (Fig. 5). Relatively high concentrations of $SO_4^{2-}$ and $NO_3^-$ were also found over the SCS. $NH_4^+$ had the lowest concentrations

among the major ions in most marine atmospheric aerosols, suggesting that little ammonia transport to the open ocean, such as Yongxing Island. However, there were some exceptions. For example, the southern Atlantic and Mediterranean Sea had the highest $NH_4^+$ concentrations among major ions (Fig. 5). Over most seas, the order was $Ca^{2+} > Mg^{2+} > K^+$. However, we found that $Mg^{2+}$ had higher concentrations than $Ca^{2+}$ in some remote ocean areas,



such as in the Pacific, Atlantic and Southern oceans. This indicates that $Ca^{2+}$ of crustal origin was difficult to

transport to the remote oceans, and $Mg^{2+}$ may mainly be from sea salt over the open ocean (Moody et al., 2014).

## 3.2 Seasonal patterns of aerosol chemical species over SCS and adjacent areas

### 3.2.1 Seasonal characteristics at Yongxing Island

As illustrated in Figs. 4 and 6, seasonal and monthly TSP concentrations and major inorganic water-soluble

ion concentrations had distinctive features at Yongxing Island. Generally, concentrations of TSP and major

inorganic ions were higher in the cool season than in the warm season (Fig. 6). Seasonal variations were the same

as those in most other studies (Arakaki et al., 2014; Wang et al., 2006; Xiao and Liu, 2004; Zhang et al., 2012).

Average TSP concentrations were $114.7 \pm 82.1$, $60.4 \pm 27.0$ and $59.5 \pm 25.6$ μg/m$^3$ in the cool, warm and

transition seasons, respectively, with the highest monthly average in November 2014 and the lowest in April (39.4

μg/m$^3$) and September (39.9 μg/m$^3$) of that year (Fig. 6). There were lower concentrations in the warm season than

in the cool season, because rainfall at Yongxing Island concentrates in the former season (Fig. 2), being the same

as many other studies (Wang et al., 2006; Zhao et al., 2015). However, there was no relationship between TSP

concentration and rainfall ($p > 0.05$; Table 1), indicating that rainfall is not a major factor controlling seasonal

variation of that concentration. The positive correlation between TSP concentration and wind speed ($p < 0.01$)

shown in Table 1 suggests that relatively strong speeds can produce many particles from both sea spray and

terrigenous matter. We discovered negative correlations between TSP concentration and temperature ($p < 0.01$)

and relatively humidity ($p < 0.01$) (Table 1), indicating that warm temperatures and high relatively humidity

enhance particle wetting and interaction. Figure S1 also shows that meteorological parameters affected the major

ions, with wind speed having a positive influence and relatively humidity, temperature and rainfall a negative one.

Based on the arrow lengths in the figure, rainfall had less effects on major ions than others.

As shown in Figs. 4 and 6, sea salt ions Na$^+$ and Cl$^-$ were characterized by a gradual increase from the transition

to cool season. Their concentrations (in μg/m$^3$) in the cool, warm and transition seasons were $4.91 \pm 1.82$ and $3.04$

$\pm 1.08$, $2.28 \pm 1.35$ and $9.93 \pm 6.78$, and $5.25 \pm 2.63$ and $3.73 \pm 3.63$, respectively, with corresponding contributions

of 52%, 57% and 57% to total major ions in those seasons. The highest Na$^+$ and Cl$^-$ concentrations in a single

sample were found in November, with the lowest concentrations in May and April, respectively. The highest

average monthly concentrations were in November. Positive relationships between Na$^+$ or Cl$^-$ and wind speed in

Table 1 ($p < 0.01$, correlation coefficient $R = 0.44$ and $p < 0.01$, $R = 0.43$, respectively) and small angles ($<< 90^o$)

between Na$^+$ or Cl$^-$ and wind speed (long arrows) in Fig. S1 at Yongxing Island suggest that sea salt concentrations



were dependent on wind speed. This is consistent with results at Chichijima Island (Boreddy and Kawamura, 2015).

There was a negative relationship between $Na^+$ and rainfall ($p < 0.05$) but no relationship between $Cl^-$ and rainfall ($p > 0.05$), showing that $Na^+$ mainly existed in coarse particles and was readily removed by rainfall (Fig. S1). As shown in Table 1 and Fig. S1, concentrations of $Na^+$ and $Cl^-$ were also negatively influenced by temperature and relatively humidity.

As shown in Fig. 6, the highest monthly average concentrations of $Ca^{2+}$ were in February. Its monthly trends

were different from those of TSP, $Na^+$ and $Cl^-$, suggesting that $Ca^{2+}$ from terrestrial dust sources may be influenced by different factors. $Ca^{2+}$ accounted for 10%, 13% and 8% of total major ions in the cool, transition and warm seasons, respectively (Fig. 4). There was no correlation between $Ca^{2+}$ and wind speed, in contrast with TSP, $Na^+$ and $Cl^-$ (Table 1). However, there was a negative relationship between $Ca^{2+}$ and rainfall ($p < 0.05$; Table 1 and Fig. S1). These results suggest that $Ca^{2+}$ existed in coarse particles that can be readily removed by rainfall, the same as

$Na^+$. Thus, a low mass concentration was observed for $Ca^{2+}$ in the rainy (warm) season (Fig. 6), with a low percentage being in the warm season in Fig.4.

Although $Mg^{2+}$ is often treated as crustal-derived ions and elements in continental studies (Zhang et al., 2015), its highest monthly average concentrations were in November at Yongxing Island, the same as $Na^+$ and $Cl^-$ (Fig. 5). As shown in Fig. 5 and Table 1, similar trends and strong correction were observed among $Na^+$, $Cl^-$ and $Mg^{2+}$,

suggesting that $Mg^{2+}$ may mainly derive from sea salt rather than continental sources. However, there were no relationships between $Mg^{2+}$ and wind speed, temperature, relatively humidity, or rainfall (Table 1), in contrast to other ions, such as $Na^+$ and $Cl^-$. These results reveal that $Mg^{2+}$ has complex sources or behaviors in the marine atmosphere at Yongxing Island. The same phenomenon was found in rainwater (Xiao et al., 2016). Moody et al. (2014) suggested that $Na^+$ may exist in super-micron size aerosols, whereas $Mg^{2+}$ in sub-micron size aerosols are

slightly influenced by meteorological parameters.

As a tracer for biomass burning, $K^+$ concentrations were $0.42 \pm 0.23$ μg/m$^3$ in the cool season, $0.22 \pm 0.18$ μg/m$^3$ in warm season, and $0.15 \pm 0.07$ μg/m$^3$ in the transition season at Yongxing, with the maximum monthly average concentrations in February and the minimum in July (Fig. 6). However, the lowest nss-$K^+$ monthly average concentration was in August. The results show that nss-$K^+$ is derived from Chinese biomass/biofuel burning in the

cool season (Lawrence and Lelieveld, 2010). Streets et al. (2003) computed that China contributes 25% of total biomass burning in Asia. Many sites in Chinese coastal regions had higher $K^+$ and nss-$K^+$ concentrations than those at Yongxing Island (Wang et al. 2006), further indicating that Chinese and other Northeast Asian regions' biomass/biofuel burning have a strong influenc on atmospheric composition over the SCS.




Biomass/biofuel burning releases not only $K^+$ but also $SO_2$ and $NO_x$ (Lawrence and Lelieveld, 2010).

Furthermore, considerable fossil fuel burning and industrial processes generate large amounts of $SO_2$ and $NO_x$ in northern Asia (Lawrence and Lelieveld, 2010), resulting in the transport of substantial secondary inorganic aerosols containing $SO_4^{2-}$ and $NO_3^-$ to the SCS. Therefore, similar to $K^+$, the highest monthly concentrations of $SO_4^{2-}$ and $NO_3^-$ were observed in February, being at $13.08 \pm 9.04$ and $4.99 \pm 4.33$ µg/m$^3$, respectively (Fig. 6). As shown in the figure, $SO_4^{2-}$ concentrations in the cool and warm seasons were $7.22 \pm 3.92$ and $3.26 \pm 1.26$ µg/m$^3$, respectively,

accounting for 26% and 22% of total major ions, and $NO_3^-$ concentrations were $2.43 \pm 1.54$ and $1.30 \pm 0.64$ µg/m$^3$, accounting for 9% and 9%. $NO_x$ emission from fossil fuel combustion made up 61% and 76% of total $NO_x$ emission in southern and northern Asia, respectively, and $SO_2$ from the same source made up 77% and 75% (Lawrence and Lelieveld, 2010). According to the aerosol $\delta^{15}N$-$NO_3^-$ at Yongxing Island, $NO_3^-$ mainly came from coal combustion in China during the cool season, and from natural emissions during the warm season (Xiao et al., 2015). In recent

years, $NO_x$ emission has also increased greatly because of increasing energy demand, although coal-fired power plants have been restricted (Zhao et al., 2015).

The $NH_4^+$ showed maxima in the cool season and minima in the warm season, being $0.08 \pm 0.08$ and $0.04 \pm 0.03$ µg/m$^3$, respectively. Atmospheric $NH_x$ is usually rapidly deposited near source regions and has a short residence time, about several hours in the marine boundary layer (Boreddy and Kawamura, 2015; Xiao et al., 2012a;

Xiao and Liu, 2002). Thus, $NH_x$ transportation from continental to remote sea sites is difficult. Therefore, $NH_4^+$ in aerosol at Yongxing Island was possibly from marine biogenic emissions, as being reported at other marine sites (Altieri et al., 2014; Jickells et al., 2003).

### 3.2.2 Seasonal patterns over SCS and adjacent areas

The spatial variability in seasonal patterns of the major inorganic ionic components at Yongxing Island and adjacent sites of Acid Deposition Monitoring Network in East Asia (EANET) is portrayed in Fig. 7. In general, total major inorganic ionic concentrations tended to be higher in cool seasons and lower in warm seasons to the north of Phnom Penh, including Phnom Penh, Hoa Binh, Hanoi, Hongwen, Hedo, Ogasawara, and Yongxing Island, consistent with previous studies (Boreddy and Kawamura, 2015; Wang et al., 2006; Xiao and Liu, 2004). There

was no substantial seasonal variation at other sites of EANET, and there was no strong seasonal variation of rainfall there either. These results suggest that rainfall and wind patterns influence the ionic seasonal variations (Lawrence and Lelieveld, 2010; Wang et al., 2006; Xiao et al., 2013; Xiao and Liu, 2004). Additionally, total major ionic concentrations were higher in the north than in the south, indicating more anthropogenic pollutants in the north,



such as $SO_4^{2-}$ and $NO_3^-$ (Lawrence and Lelieveld, 2010). As it is well known, the most densely populated regions

in the north, including Hanoi, northeastern China, Pearl River Delta of China, Korea, and Japan release large

amounts of pollutants (Lawrence and Lelieveld, 2010), which then transport to the SCS in the cool seasons (Fig.

1).

The total ionic concentrations were higher at the three islands than sites to the south of Phnom Penh. As shown

in Fig. 7, relatively high concentrations of $Na^+$ and $Cl^-$ were found at those islands, suggesting that ions from sea

salt had large contributions to total major ions, i.e., 52.8%, 62.5% and 55.6% at Yongxing, Ogasawara and Hedo,

respectively. This represents high mass concentrations of sea salt in the marine atmospheric aerosol. The highest

concentrations of both $Na^+$ and $Cl^-$ appeared in November at Yongxing and Hedo islands, which were influenced

by a strong northeast monsoon. The highest concentrations of both $Na^+$ and $Cl^-$ were in September at Ogasawara

Island, which were influenced by a strong southeast monsoon from the Pacific. The relationship between $Na^+$, $Cl^-$

and wind speed at Yongxing ($p < 0.01$) is shown in Table 1 and Fig. S1. Other sites in Fig. 7 were also influenced

by wind speed and winds directly from the ocean. However, the highest $Na^+$ and $Cl^-$ concentrations at some sites

did not appear in the same month, e.g., at Hongwen, the highest concentrations of $Na^+$ were in April and the highest

of $Cl^-$ were in January. Excess $Cl^-$ in January there may have come from biomass burning and coal combustion in

China (Duan et al., 2006).

The highest concentrations of $Mg^{2+}$ were in the same months as $Na^+$ at most sites, indicating that $Mg^{2+}$ may

be from sea salt with $Na^+$. The exceptions were at Hoa Binh and Tanah Rata with the maximum $Mg^{2+}$ concentrations

being in December and July, respectively. This suggests that $Mg^{2+}$ originates from the crust rather than oceans

(Xiao and Liu, 2004), or from both crust and oceans at these sites. Hoa Binh was influenced by the northeast

monsoon, which carries strongly weathering crustal matters from China Yunnan-Guizhou Plateau karst (Hien et

al., 2004; Xiao et al., 2013), and there was a strong relationship between $Mg^{2+}$ and $Ca^{2+}$ ($R = 0.7$, $p < 0.05$). $Ca^{2+}$,

a tracer for dust, had its highest concentrations in July at Phnom Penh, Tanah Rata, Petaling Jaya, Serpong, and

Danum Valley, all of which are located in the south of the SCS. In these regions, relatively little rainfall (rainfall

data from EANET) and strong sunlight were observed in that month, leading to strong weathering that generated

$Ca^{2+}$. However, the highest $Ca^{2+}$ concentrations were found at other sites in the cool season, during which there

was much dust from Northeast Asia (Fig. S2; Boreddy and Kawamura, 2015; Liu et al., 2014; Wang et al., 2011).

This result is consistent with earlier studies (Boreddy and Kawamura, 2015; Cheng et al., 2000; Liu et al., 2014;

Zhao et al., 2015). The $Ca^{2+}$ data also proved that Asian dust can affect the northern SCS, but it is difficult for Asian

dust to be transported to the southern SCS (Figs. 1 and S2).



Figure 8 shows fire spot data from MODIS global fire mapping around the SCS during March 2014 through

February 2015. Additionally, dynamic smoke surface concentrations every day in that period and region is shown

in Fig. S2. The fire spot and smoke data give information on seasonal variations of biomass burning around the

SCS. This activity was strong from January to April in the west of the SCS, including Vietnam, Thailand and Laos,

and between July and October in the south of the SCS, including Malaysia and Indonesia (Figs. 8 and S2). These

data are consistent with other studies showing substantial monthly CO emissions from biomass burning during

February–April and August–October in Southeast Asia, and February–May in southern China and Taiwan (Streets

et al., 2003). $K^+$ is commonly used as a tracer of biomass and biofuel burning (Deng et al., 2010). As shown in Fig.

7, we found that the maximum $K^+$ was in the aforementioned months at most sites, suggesting that Asian biomass

burning heavily influenced the SCS region.

Biomass burning is an important source of atmospheric pollutants, such $SO_2$, $NO_x$, and $NH_3$ (Streets et al.,

2003). According to the reported by Streets et al. (2003), emissions of biomass burning in Asia contribute 0.37 Tg

of $SO_2$, 2.8 Tg of $NO_x$, and 0.92 Tg of $NH_3$, or 1.1%, 11% and 3.3% of total Asian emissions, respectively. Natural

emissions include $SO_2$ and $NH_3$ from marine and soil biological processes (Altieri et al., 2014; Boreddy and

Kawamura, 2015; Xiao et al., 2012a), and $NO_x$ from those processes and lightning (Price et al., 1997; Xiao et al.,

2015). Certainly, fossil fuel combustion, industrial processes, biofuel burning, agricultural and waste handling also

generate large quantities of $SO_2$, $NO_x$, $NH_3$ in Asia (Lawrence and Lelieveld, 2010; Liu et al., 2013; Xiao et al.,

2012a; Xiao et al., 2014; Xiao et al., 2015). In general, the three marine sites (Yongxing, Hedo, and Ogasawara

islands) had smaller proportions of $SO_4^{2-}$, $NO_3^-$ and $NH_4^+$ than inland sites, with the three ions accounting for ~35%

at the marine sites and up to 65% at the other sites. This indicates that anthropogenic contributions are smaller over

remote open oceans than at continental sites. Figure 6 shows that the highest $SO_4^{2-}$, $NO_3^-$ and $NH_4^+$ concentrations

were found during the cool season in the north of Phnom Penh, including Phnom Penh, Hoa Binh, Hanoi, Hongwen,

Hedo, Ogasawara, and Yongxing, consistent with total inorganic major ions. This indicates that the pollutants from

Northeast Asia have a great impact on the Northwest Pacific. Figure S2 confirms these findings. We also found

that most sites in the south of Phnom Penh had maximum $SO_4^{2-}$ and $NO_3^-$ concentrations in the same months as the

highest $K^+$ concentrations occurred, suggesting that biomass and biofuel burning are important sources for $SO_4^{2-}$

and $NO_3^-$ in those regions. Lawrence and Lelieveld (2010) found that such burning was important in the emissions

of $SO_4^{2-}$ and $NO_3^-$ in southern Asia, whereas fossil fuel combustion and industrial processes tended to be dominant

in northern Asia (Xiao et al., 2015). However, maximum $NH_4^+$ concentrations at some sites (e.g., Petaling Jaya,

Serpong, Danum Valley) were inconsistent with $SO_4^{2-}$ and $NO_3^-$. Moreover, there was no relationship between



$SO_4^{2-}$ or $NO_3^-$ and $NH_4^+$ at these sites in the south of Phnom Penh, including Phnom Penh and Yongxing Island

(both $p > 0.05$). The results are inconsistent with previous studies (Boreddy and Kawamura, 2015; Hsu et al., 2007; Wang et al., 2006; Xiao et al., 2013; Xiao and Liu, 2004). In the marine atmosphere, $H_2SO_4$ and $HNO_3$ can react with NaCl to generate $Na_2SO_4$, $NaNO_3$ in coarse particles, and HCl (Boreddy and Kawamura, 2015; Xiao et al., 2015). $NH_4^+$ is often predominant in fine particles and may exist in the form of $(NH_4)_2SO_4$ in their accumulation (Ooki et al., 2007; Ottley and Harrison, 1992). However, ratios of $NH_4^+$ to nss-$SO_4^{2-}$ in the size range D > 0.22 μm

(D = diameter) decreased with particle size (Ooki et al., 2007). Ooki et al. (2007) found that in the range 0.06 < D < 0.22 μm, $(NH_4)_2SO_4$ was mainly derived from marine biogenic sources. In addition, ammonium salts such as $NH_4Cl$ and $NH_4NO_3$ are readily dissociable by evaporation in the marine atmosphere (Ottley and Harrison, 1992), which caused a weak relationship between $NH_4^+$ and other ions at Yongxing Island.

## 370   3.3 Aerosol chemical principal component analysis

3.3.1 Correlation analysis

Correlation analysis was used to characterize relationships among the ions and distinguish potential sources of ionic constituents (Xiao et al., 2013; Xiao and Liu, 2004). As shown in Table 1, $p$-values among all ions except $NH_4^+$ were < 0.01 at Yongxing Island, indicating that the correlation between each two ions was statistically

significant. $R$ between $Na^+$ and $Cl^-$ was > 0.8, suggesting that they have a major common source and may exist in NaCl within aerosols in the marine atmosphere. The mole equivalent ratios of $Cl^-/Na^+$ (neq/L) in aerosols were slightly larger than seawater in annual, cool and warm seasons at Yongxing Island (Table 2). This suggests that $Cl^-$ enrichment had an anthropogenic or natural biomass burning origin (Duan et al., 2006; Jung et al., 2012; Xiao et al., 2013). However, as shown in Table 2, the ratio of $Cl^-/Na^+$ was slightly smaller than seawater in the transition

season, most likely because air masses were primarily from local areas far from the continent (Fig. 1), where wind is weak (Fig. 2). Thus, $Cl^-$ depletion occurred through the volatilization of HCl during the reaction of NaCl with $N_xO_y$, $HNO_3$, or $H_2SO_4$ (Hsu et al., 2007; Jung et al., 2012; Xiao et al., 2015). Nevertheless, most $Na^+$ originates from sea salt, but part is from dust, fossil combustion, and biomass burning (Tiwari et al., 2013; Xiao et al., 2016; Zhang et al., 2015). There was a strong relationship between $K^+$ and $Cl^-$ (Table 1), indicating that KCl is partly

from sylvite weathering and biomass burning (Xiao et al., 2013). In addition, $K^+$ had strong correlation with $Ca^{2+}$, implying that they may come from crustal components. According to the ratio of $K^+/Na^+$ at Yongxing Island and in seawater (Table 3), a part of $K^+$ is from seawater. Further, a strong relationship between $K^+$ and $SO_4^{2-}$ was found (Table 1), indicating that they have a common source, i.e., biomass burning (Li et al., 2003; Streets et al., 2003).





There was also a strong relationship between $Ca^{2+}$ and $SO_4^{2-}$, meaning that they may exist in the form of $CaSO_4$ in

the marine atmosphere. The secondary aerosols $SO_4^{2-}$ and $NO_3^-$ were also strongly correlated, which may be attributed to similarity of their chemical behavior and a common source of their precursors $SO_2$ and $NO_x$ (Xiao et al., 2013). Ratios of $Mg^{2+}/Na^+$ were equal to those of seawater throughout the year at Yongxing (Table 2), indicating that most $Mg^{2+}$ was likely from sea salt. However, $R$ between $Mg^{2+}$ and $Na^+$ was much smaller than that between $Cl^-$ and $Na^+$, suggestive of a different behavior of $Mg^{2+}$ to that of $Na^+$. The ratios of $NO_3^-/nss$-$SO_4^{2-}$ and $NH_4^+/nss$-

$Ca^{2+}$ were about 0.7 and 0.07 at Yongxing (Table 3), respectively. This may be because that nss-$SO_4^{2-}$ and nss-$Ca^{2+}$ were the major components from the continent, and demonstrated the presence of $CaSO_4$. There were large $R$ among most ions (Table 2), suggesting that they existed in the following forms: $NaCl$, $CaCl_2$, $MgCl_2$, $KCl$, $Na_2SO_4$, $CaSO_4$, $MgSO_4$, $K_2SO_4$, $(NH_4)_2SO_4$, $NaNO_3$, $Ca(NO_3)_2$, $Mg(NO_3)_2$, $KNO_3$, and $NH_4NO_3$.

3.3.2 Principal component analysis and classical multidimensional scaling

Principal component analysis (PCA) was also used to explore the relationship among the aerosol ions at Yongxing Island (Fig. 9a). As seen in the figure, the first two components (PC1 and PC2) explained 96.0% of the variance in total and 84.5% and 11.5% individually. The first component captured the variance of $Cl^-$, $Na^+$, $Ca^{2+}$, $K^+$, $Mg^{2+}$, $SO_4^{2-}$ and $NO_3^-$, indicating that they had common sources or chemical behaviors, as mentioned above.

The angle between two ions and arrow lengths reflect their relationship and the contribution of different ions to the principal component. As shown in Fig. 9a, the samples from the cool season had a greater contribution to PC1 than in other seasons.

To better classify the ions, classical multidimensional scaling (CMDS) of the correlation coefficients based on Table 2 is plotted in Fig. 9b. As shown in that figure, in the same quadrant, they may have common sources or

behaviors. For example, $Na^+$ and $Cl^-$ in the same quadrant indicates that they possibly derived from sea salt. $SO_4^{2-}$ and $Ca^{2+}$ in the same quadrant implies that they likely existed in the form of $CaSO_4$ in the marine atmosphere. Although $Mg^{2+}$ was alone in a quadrant, it was in the lower right quadrant with $Na^+$ and $Cl^-$, suggesting that they had a common source from sea salt, but different behaviors. Previous study reported that $Na^+$ may exist in super-micron size aerosol, and $Mg^{2+}$ in sub-micron size aerosol (Moody et al., 2014). $SO_4^{2-}$, $Ca^{2+}$, $K^+$, $Cl^-$ and $Na^+$ in the

positive Dimension 1 indicates that they may exist in coarse particles. It was clear that $NH_4^+$ is widely separated from other ions, indicating that it was from a unique source.

3.3.3 Source identification and apportionment



Based on the PMF model, five potential sources of atmospheric chemical components at Yongxing Island
were identified: sea salt, crust, biomass burning, fossil fuel combustion, and marine biogenic. Table 3 summarizes
source apportionment of the relative contributions of each identified source to the TSP and major ions. These
sources have average contributions of 26% for sea salt and 53% for crust to TSP, and < 10% for others.

The first source, sea salt, generally has strong marine elements, such as $Na^+$ and $Cl^-$. They contributed 74%
($Na^+$) and 82% ($Cl^-$) from sea salt at Yongxing Island (Table 3). Other significant sources were crust, biomass
burning, fossil fuel combustion and marine biogenic, with contributions < 10% for $Na^+$ and $Cl^-$. According to data
of rainwater at Yongxing Island, a part of $Na^+$ and $Cl^-$ could be from crust and produced by burning (Xiao et al.,
2016). Coal combustion and biomass burning also produce $Na^+$ and $Cl^-$ (Liu et al., 2000; Tiwari et al., 2013; Zhang
et al., 2015). Zhang et al. (2015) found that coal combustion was the most likely dominant source of $Cl^-$ in Beijing.
The mole equivalent $Cl^-/Na^+$ ratios were larger in the cool season than in the transition and warm seasons at
Yongxing, indicating that the crust, fossil combustion, and biofuel and biomass burning affected $Na^+$ and $Cl^-$
concentrations over the Northwest Pacific (Figs. 8 and S2). As shown in Table 2, there were strong relationships
between $Na^+$ ($Cl^-$) and $SO_4^{2-}$ ($Ca^{2+}$, $K^+$), further proving that crust, fossil combustion and biomass burning can
generate $Na^+$ and $Cl^-$. Moreover, NaCl can react with acids such as $H_2SO_4$, $HNO_3$ and $H_2C_2O_4$, altering $Na^+$ and $Cl^-$
concentrations in the marine atmosphere and producing secondary chlorine-containing salt (Boreddy and
Kawamura, 2015). Sea salt provided $K^+$, $Mg^{2+}$ and $SO_4^{2-}$, constituting 42%, 63% and 31%, respectively. The results
are consistent with other studies (Boreddy and Kawamura, 2015)

The second source, crust, has a substantial crustal element $Ca^{2+}$, which is a tracer of crust (Suzuki and Tsunogai,
1988; Xiao et al., 2013; Xiao and Liu, 2004). Absolutely, $Ca^{2+}$ mainly came from the crust, which had a contribution
of 50% (Table 3). A strong relationship between TSP and $Ca^{2+}$ observed at Yongxing Island ($R$=0.92, $p$<0.01, Table
1) indicated that crust had a large contribution to TSP (53%; Table 3). Biomass burning and fossil combustion also
had contributions (9% and 18%, respectively) to $Ca^{2+}$. $CaSO_4$ and sulfate containing both K and Ca have been
observed in smoke and have been reported to originate from biomass burning (Allen and Miguel, 1995; Li et al.,
2003). This implies that Asian biomass and biofuel burning increased $Ca^{2+}$ concentrations of aerosols in the marine
atmosphere. Studies have found that atmospheric $Ca^{2+}$ also derives from fossil fuel combustion (Hutton and Symon,
1986). Some $Ca^{2+}$ may be affected by local coral (Suzuki and Tsunogai, 1988; Xiao et al., 2016). There is a large
percentage of live coral cover (Huang et al., 2006). Large corals have been dying with rapid development of the
Xisha islands (including Yongxing), which have generated large amounts of $CaCO_3$ and become a source of $Ca^{2+}$
in marine atmospheric aerosols (Xiao et al., 2014; Xiao et al., 2016).





The third source, biomass burning, is characterized by substantial $K^+$, which is an effective tracer of biomass
and biofuel burning aerosols (Zhang et al., 2015), with a contribution of 41% from such burning at Yongxing Island.
$K^+$ salts and organic particles are the dominant species in smoke, with more KCl in young smoke, and $K_2SO_4$ and
$KNO_3$ in aged smoke from biomass burning (Li et al., 2003). As shown in Figs. 8 and S2, biomass burning was
more frequent in the cool season than in the warm and transition seasons over the Northwest Pacific, yielding ratios
of $K^+/Na^+$ (neq/L) that were larger in the cool season (Table 2). Another main source for $K^+$ was sea salt, with a
contribution of 42% (Table 3). In addition, biomass burning produces $SO_2$ and $NO_x$ (Streets et al., 2003; Xiao et
al., 2015; Zhang et al., 2015); they accounted for 7% of $SO_4^{2-}$ and 1% of $NO_3^-$.

The fourth source is fossil fuel (especially coal) combustion, which releases large amounts of $SO_2$, $NO_x$ and
$NH_3$ (Xiao et al., 2012b; Xiao et al., 2014; Pan et al., 2016). Accordingly, fossil combustion contributed 22%, 56%
and 26% to $SO_4^{2-}$, $NO_3^-$ and $NH_4^+$ at Yongxing Island, respectively. $SO_4^{2-}$ and $NO_3^-$ emissions from fossil
combustion may originate from Northern Asia in the cool season (Lawrence and Lelieveld, 2010; Xiao et al., 2015).
In Chinese coastal provinces, emission intensities of $SO_2$ and $NO_x$ were about 10 and 15 tons/km$^2$, respectively,
much higher than other Chinese provinces (China Environment Statistical Yearbook, 2014). These results indicate
that human activities clearly affected marine atmospheric chemical compositions.

The fifth source is marine biogenic, which released $NO_x$, $NH_3$ and DMS (Altieri et al., 2014; Jickells et al.,
2003; Boreddy and Kawamura, 2015; Phinney et al., 2006), with respective contributions of 19% to $NO_3^-$, 69% to
$NH_4^+$, and 38% to $SO_4^{2-}$ at Yongxing Island (Table 3). Substantial $NH_x$ may be released from degraded organic
nitrogen-containing compounds and excretion from zooplankton in the ocean (Norman and Leck, 2005). The
contribution of marine biogenic sources to $NH_4^+$ was much larger at Yongxing than at global marine atmospheric
$NH_x$ sources in the review of Duce et al. (2008), which showed 87.5% from anthropogenic sources. However,
Altieri et al. (2014) found that the anthropogenic contribution was < 87.5% at Bermuda, an island in the North
Atlantic Ocean (Fig. 5). DMS is the most abundant marine biogenic volatile sulfur emitted from the ocean surface
to atmosphere, and can be oxidized to $SO_4^{2-}$ in the marine atmosphere (Phinney et al., 2006). Yang et al. (2015)
reported that biogenic $SO_4^{2-}$ from the Bohai and northern Yellow seas near China was 0.114–0.551 µg/m$^3$, with an
average of 0.247 µg/m$^3$, accounting for 1.4% of nss-$SO_4^{2-}$. Biogenic $SO_4^{2-}$ in the northern SCS was ~1.2 and 0.6
µg/m$^3$ in summer and winter, respectively (Zhang et al., 2007), constituting ~8% and 12% of nss-$SO_4^{2-}$. However,
large ratios of biogenic $SO_4^{2-}$ to nss-$SO_4^{2-}$ were observed at the remote Pacific islands of Oahu and Midway, at 55%
and 70% (Arimoto et al., 1996), being consistent with our results. Thus, natural sources had large contributions to
marine atmospheric aerosols over the Northwest Pacific.




### 3.4 Regional sources deduced from trajectory and CWT analyses

Air masses at Yongxing Island had obvious unique and seasonal variations, from northeast of the island in the cool season, southwest in the warm season, and southeast in the transition season (Fig. 10). This reveals that aerosol or chemical compositions at the island originated from different regions in different seasons (Figs. 1, 8, 10 and S2). CWTs were plotted for TSP and major ions (Fig. 10) to explore likely regional sources and transport pathways for the island. A major discovery was that air masses with high TSP concentrations were from China coastal regions bordering the Yellow and East China Seas and northern South China (Fig. 10). This is consistent with the seasonal variations of TSP concentrations in Fig. 6. The average aerosol optical thickness (AOT) over the Northwest Pacific (Fig. 1) confirmed this result. A relatively large average AOT was found over the northern SCS and East China Sea in the cool season, and Karimata Strait in the warm season (Fig. 1). But there was a relatively low average AOT over the entire SCS in the transition season (Fig. 1).

CWTs for $Ca^{2+}$ had a similar trend as that of TSP, larger in the cool season and lower in the warm season, indicating that dust from Northeast Asia influenced aerosol chemistry in the remote marine atmosphere (Figs. 1, 10 and S2). Although the CWT for $Mg^{2+}$ was also larger in the cool season and lower in the warm season, air masses with relatively high concentrations of $Mg^{2+}$ originated offshore of China (Fig. 10). This indicates that $Mg^{2+}$ was mainly from sea salt, consistent with the PMF results (Table 3).

CWTs for $K^+$, $SO_4^{2-}$, and $NO_3^-$ concentrations showed considerable seasonal variations, higher in the cool season and lower in the warm and transition seasons (Fig. 10). As with TSP and $Ca^{2+}$, air masses from China coastal regions had high $SO_4^{2-}$ and $NO_3^-$ concentrations in the cool season (Fig. 10), owing to rapid economic development and great coal demand in the country, especially in its coastal regions (Lawrence and Lelieveld, 2010). However, the three ions may have different sources; $K^+$ was mainly derived from biomass and biofuel burning in Northeast Asia (Fig. 8; Atwood et al., 2013; Streets et al., 2003), whereas $NO_3^-$ and $SO_4^{2-}$ were mainly from fossil fuel combustion in that region (Table 3; Xiao et al., 2014 and 2015a; Lawrence and Lelieveld, 2010). Figure S2 also shows that $SO_4^{2-}$ from central and eastern China reached coastal regions in the cool season. According to data from the China Environment Statistical Yearbook (2014), average $SO_2$, $NO_x$, and soot and dust emissions from Chinese coastal regions (Liaoning, Hebei, Shandong, Jiangsu, Zhejiang, Fujian and Guangdong provinces, and the cities of Shanghai, Beijing and Tianjin) exceeded 10, 15 and 5 tons/km$^2$, respectively, much greater than other provinces or cities in the country. In addition, higher concentrations of $SO_2$ and $NO_x$ in Northeast Asia than South Asia (Lawrence and Lelieveld, 2010) indicate that the marine atmospheric aerosol chemical compositions were





influenced by Northeast Asia, especially China coastal regions in the cool season. Rapidly growing economies and
high population densities in these regions (Kim et al., 2014) release pollutants that are transported to the Northwest
Pacific. Biogenic $SO_4^{2-}$ from the Chinese seas may affect the atmospheric aerosol chemical compositions.

As mentioned above, in Table 3, $NH_4^+$ may largely come from marine biogenic sources, and air masses with
high $NH_4^+$ concentrations were from remote open oceans such as the southeastern and northeastern SCS (Fig. 10).
This suggests that it is feasible for the ocean to be a $NH_4^+$ source. Globally, 87.5% of marine atmospheric $NH_x$ is
from human activities (Duce et al., 2008). However, there was little $NH_x$ from such activities at Yongxing Island,
consistent with Altieri et al. (2014). There were relatively higher $NH_4^+$ concentrations in the cool season and lower
values in the warm season at Yongxing Island (Fig. 6). The Northwest Pacific Ocean including the SCS has
oligotrophic surface water and is a nitrogen-limited region (Chen et al., 2004; Kim et al., 2014; Wu et al., 2001),
with only a hundred nanomoles per liter of $NH_4^+$ in surface water (Lin, 2013). Thus, it is in some ways
counterintuitive that marine atmospheric aerosol $NH_4^+$ at Yongxing Island is from marine sources. Altieri et al.
(2014) suggested that the efficient kinetics of ammonia evasion from surface seawater causes $NH_3$ to accumulate
in the marine atmosphere. $NH_3$ may be released from degraded organic nitrogen-containing compounds and
excretion from zooplankton (Norman and Leck, 2005). Lin (2013) found average $N_2$ fixation rates in the SCS of
4.9 and 0.7 μmol N/m$^3$/d in the cool season and warm seasons, respectively. He suggested that this seasonal
variation may be caused by the intensity of the Kuroshio intrusion in the SCS and Fe supply from the atmosphere
in the cool season, which enhances primary productivity. The unique temporal variations of $N_2$ fixation rates in the
SCS are consistent with our CWT results. Moreover, as shown in Figs. 1 and S2, dust was transported to remote
open oceans in the cool season, likely containing substantial Fe (Wu et al., 2001). Increased Fe would greatly
increase productivity in these marine regions (Wu et al., 2001).


**4 Conclusions**

Chemical compositions of 1-year aerosols at Yongxing Island were investigated to help better understanding
of their chemical characteristics, sources, and transport pathways over the SCS. Sea salt ($Na^+$ and $Cl^-$) had the
greatest contribution to total major inorganic ions in aerosols at the island, followed by $SO_4^{2-}$, $Ca^{2+}$, $NO_3^-$, $Mg^{2+}$, $K^+$
and $NH_4^+$. The concentrations of TSP and all major inorganic ions showed seasonal variations, higher in the cool
season and lower in the warm season, which was influenced by meteorological parameters (e.g., wind speed,
temperature, relative humidity and rainfall) and air masses. Using PMF and CWT models, fire spot and AOT, we
found that $Na^+$, $Cl^-$, and $Mg^{2+}$ were mainly derived from sea salt, $Ca^{2+}$ from soil dust, $K^+$ from biomass burning,



and $NO_3^-$ from fossil fuel combustion (especially coal combustion in the Northern Asia). $SO_4^{2-}$ was from marine
biogenic sources, sea salt, and fossil fuel combustion. Surprisingly, $NH_4^+$ was mainly from marine biogenic
processes in the remote ocean. In summary, Asian dust, biomass burning and fossil fuel combustion seriously
affected marine atmospheric aerosol chemical compositions over the Northwest Pacific.

**Acknowledgements** This work was supported by the National Natural Science Foundation of China (Grant nos.
41425014, 41663003, and 41203015), Strategic Priority Research Program of the Chinese Academy of Sciences
(Grant nos. XDA11030103 and XDA11020202), Doctoral Scientific Research Foundation of East China University
of Technology (Grant no. DHBK2015327), and Scientific Research Foundation of East China University of
Technology for Science and Technology Innovation Team (Grant no. DHKT2015101).

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

**Table Captions**

**Table 1** Correlation coefficients among major ions in aerosol and meteorological parameters.

**Table 2** Mole equivalent ratios for major ionic species in aerosols at Yongxing Island (annual, cool, transition and warm seasons), together with seawater ratios for comparison.

**Table 3** Relative contributions (%) for different major ions from six factors of TSP at Yongxing Island over the year, based on PMF model.

**Figure Captions**

**Figure 1** Distribution of seasonal average aerosol optical thickness (AOT) at 550 nm (T550) over Northwest Pacific in cool, warm and transition seasons during sampling period. Monthly AOT products (from Moderate Resolution Imaging Spectrometer, MODIS) with 4-km resolution were downloaded from Globcolour (http://hermes.acri.fr/). The GlobColour project began in 2005 as an ESA Data User Element project to provide a continuous dataset of merged Level 3 Ocean Colour products.



**Figure 2** Three-hour temperature, relative humidity, wind speed and precipitation at Yongxing Island during sampling period (March 2014 through February 2015).

**Figure 3** Annual average mass concentrations of aerosol chemical species at Yongxing Island. Box boundary indicates 25th and 75th percentile. Lines within the box show the mean. Whiskers above and below the box indicate 90th and 10th percentiles.

**Figure 4** Mean contributions of each major ionic component to total ionic mass concentration of (a) Yongxing Island (YXI) annual, (b) YXI cool season, (c) YXI warm season, and (d) YXI transition season.

**Figure 5** Comparisons of major ions in aerosol at Yongxing Island with global ocean. Data of Oki, Ogasawara and Hedo are from EANET (www.eanet.asia); those of Rishiri Island are from Okuda et al. (2006); those of Hawaii are from Carrillo et al. (2002); those of Bermuda are from Moody et al. (2014); those of Amsterdam Island are from

Claeys et al. (2010); those of the Arabian Sea and Indian Ocean are from Kumar et al. (2008); those of Helgoland are from Ebert et al. (2000); those of the Mediterranean Sea, northern Atlantic-1 and 2, Pacific and southern Atlantic are from Zhang et al. (2010). Pentagrams represent sampling sites on islands; others represent cruises. N.A. indicates no data.

**Figure 6** Seasonal variations of TSP mass concentration and associated species, including $Na^+$, $Cl^-$, $SO_4^{2-}$, $Mg^{2+}$,

$K^+$, $Ca^{2+}$, $NH_4^+$, and $NO_3^-$ at Yongxing Island (cool season: C; warm season: W; annual: A). Shown are the mean and standard deviation for each bar.

**Figure 7** Comparison of aerosol chemical species between Yongxing Island and around the South China Sea (data from EANET).

**Figure 8** Fire spot data from MODIS global fire mapping from March 2014 to February 2015 around South China

Sea (https://firms.modaps.eosdis.nasa.gov/firemap/).

**Figure 9** Plots of (a) principal component analysis (PCA) and (b) classical multidimensional scaling (CMDS) of correlation coefficients among major ions in aerosol samples.

**Figure 10** Ten-day back trajectories of warm (black, June through September 2014), cool (blue, March and April 2014 and October 2014 through February 2015) seasons, and transition season (red, May 2014) at Yongxing Island.

Additionally, CWT (concentration weighted trajectory) plots for daily weighted-average concentrations of TSP, $Ca^{2+}$, $Mg^{2+}$, $K^+$, $SO_4^{2-}$, $NO_3^-$ and $NH_4^+$ at Yongxing Island.

**Supporting Information**

**Figure S1.** Plot from principal component analysis (PCA) for species and environmental variables.

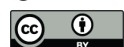



**Figure S2.** Modeled NAAPS total aerosol optical depth (AOD) for every month of March 2014 through February

2015, for total, sulfate, dust and smoke (data from http://www.nrlmry.navy.mil/aerosol/#aerosolobservations).



**Table 1** Correlation coefficients among major ions in aerosol and meteorological parameters.

|  | TSP | Na⁺ | Cl⁻ | SO₄²⁻ | Ca²⁺ | Mg²⁺ | K⁺ | NH₄⁺ | NO₃⁻ | WS | T | RH | R |
|---|---|---|---|---|---|---|---|---|---|---|---|---|---|
| TSP | 1 | 0.77** | 0.92** | 0.77** | 0.92** | 0.32** | 0.75** | -0.05 | 0.52** | 0.36** | -0.47** | -0.44** | -0.20 |
| Na⁺ |  | 1 | 0.91** | 0.69** | 0.72** | 0.57** | 0.78** | -0.03 | 0.48** | 0.44** | -0.51** | -0.46** | -0.27* |
| Cl⁻ |  |  | 1 | 0.71** | 0.83** | 0.49** | 0.77** | -0.04 | 0.45** | 0.43** | -0.37** | -0.36** | -0.19 |
| SO₄²⁻ |  |  |  | 1 | 0.86** | 0.56** | 0.85** | 0.26* | 0.87** | 0.04 | -0.56** | -0.58** | -0.29* |
| Ca²⁺ |  |  |  |  | 1 | 0.36** | 0.81** | -0.03 | 0.69** | 0.24 | -0.51** | -0.53** | -0.27* |
| Mg²⁺ |  |  |  |  |  | 1 | 0.63** | 0.45** | 0.59** | 0.04 | -0.12 | -0.08 | -0.18 |
| K⁺ |  |  |  |  |  |  | 1 | -0.18 | 0.72** | 0.15 | -0.51** | -0.45** | -0.27* |
| NH₄⁺ |  |  |  |  |  |  |  | 1 | 0.36** | -0.13 | -0.18 | 0.11 | -0.18 |
| NO₃⁻ |  |  |  |  |  |  |  |  | 1 | -0.05 | -0.50** | -0.50** | -0.31** |

**Correlation significant at 0.01 level (2-tailed), * significant at 0.05 level (2-tailed). WS: wind speed (m/s); T: temperature (°C); RH: relative humidity (%); R: rainfall (mm/h)





**Table 2** Mole equivalent ratios for major ionic species in aerosols at Yongxing Island (annual,

cool, transition and warm seasons), together with seawater ratios for comparison.

| | Yongxing Island | | | | Seawater[a] |
|---|---|---|---|---|---|
| | annual | cool | transition | warm | |
| $Cl^-/Na^+$ | 1.25 | 1.31 | 1.06 | 1.12 | 1.17 |
| $Mg^{2+}/Na^+$ | 0.21 | 0.21 | 0.19 | 0.23 | 0.22 |
| $K^+/Na^+$ | 0.048 | 0.051 | 0.040 | 0.042 | 0.021 |
| $Ca^{2+}/Na^+$ | 0.62 | 0.64 | 0.83 | 0.47 | 0.044 |
| $SO_4^{2-}/Na^+$ | 0.66 | 0.71 | 0.73 | 0.51 | 0.12 |
| $nss\text{-}SO_4^{2-}/Na^+$ | 0.54 | 0.58 | 0.61 | 0.39 | - |
| $NO_3^-/Na^+$ | 0.18 | 0.18 | 0.22 | 0.16 | - |
| $NH_4^+/Na^+$ | 0.022 | 0.021 | 0.044 | 0.016 | - |
| $NO_3^-/nss\text{-}SO_4^{2-}$ | 0.34 | 0.32 | 0.36 | 0.41 | - |
| $NH_4^+/nss\text{-}Ca^{2+}$ | 0.038 | 0.035 | 0.056 | 0.038 | - |

[a]Seawater ratios from Keene et al. (1986).


**Table 3** Relative contributions (%) for different major ions from potential five sources of TSP

at Yongxing Island over the year, based on PMF model.

| Source | TSP | $Na^+$ | $Cl^-$ | $K^+$ | $Ca^{2+}$ | $Mg^{2+}$ | $SO_4^{2-}$ | $NO_3^-$ | $NH_4^+$ |
|---|---|---|---|---|---|---|---|---|---|
| Sea salt | 26 | 74 | 82 | 42 | 8 | 63 | 31 | 14 | 1 |
| Crust | 53 | 1 | 10 | 7 | 50 | 2 | 2 | 10 | 3 |
| Biomass burning | 7 | 5 | | 41 | 9 | 16 | 7 | 1 | |
| Fossil fuel combustion | 7 | | 4 | | 18 | | 22 | 56 | 26 |
| Marine biogenic sources | 7 | 20 | 4 | 10 | 15 | 18 | 38 | 19 | 69 |





**Figure 1** Distribution of seasonal average aerosol optical thickness (AOT) at 550 nm (T550) over Northwest Pacific in cool, warm and transition seasons during

sampling period. Monthly AOT products (from Moderate Resolution Imaging Spectrometer, MODIS) with 4-km resolution were downloaded from Globcolour

(http://hermes.acri.fr/). The GlobColour project began in 2005 as an ESA Data User Element project to provide a continuous dataset of merged Level 3 Ocean

Colour products.

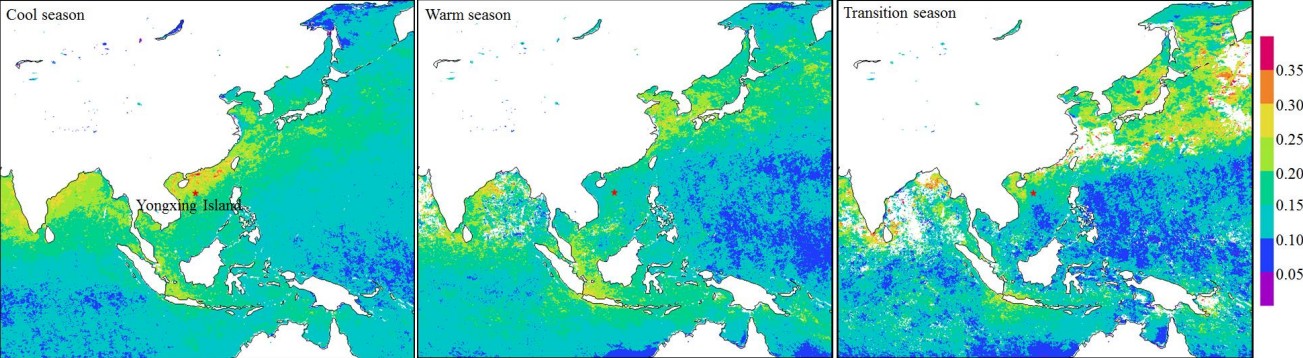





**Figure 2** Three-hour temperature, relative humidity, wind speed and precipitation at Yongxing

Island during sampling period (March 2014 through February 2015).

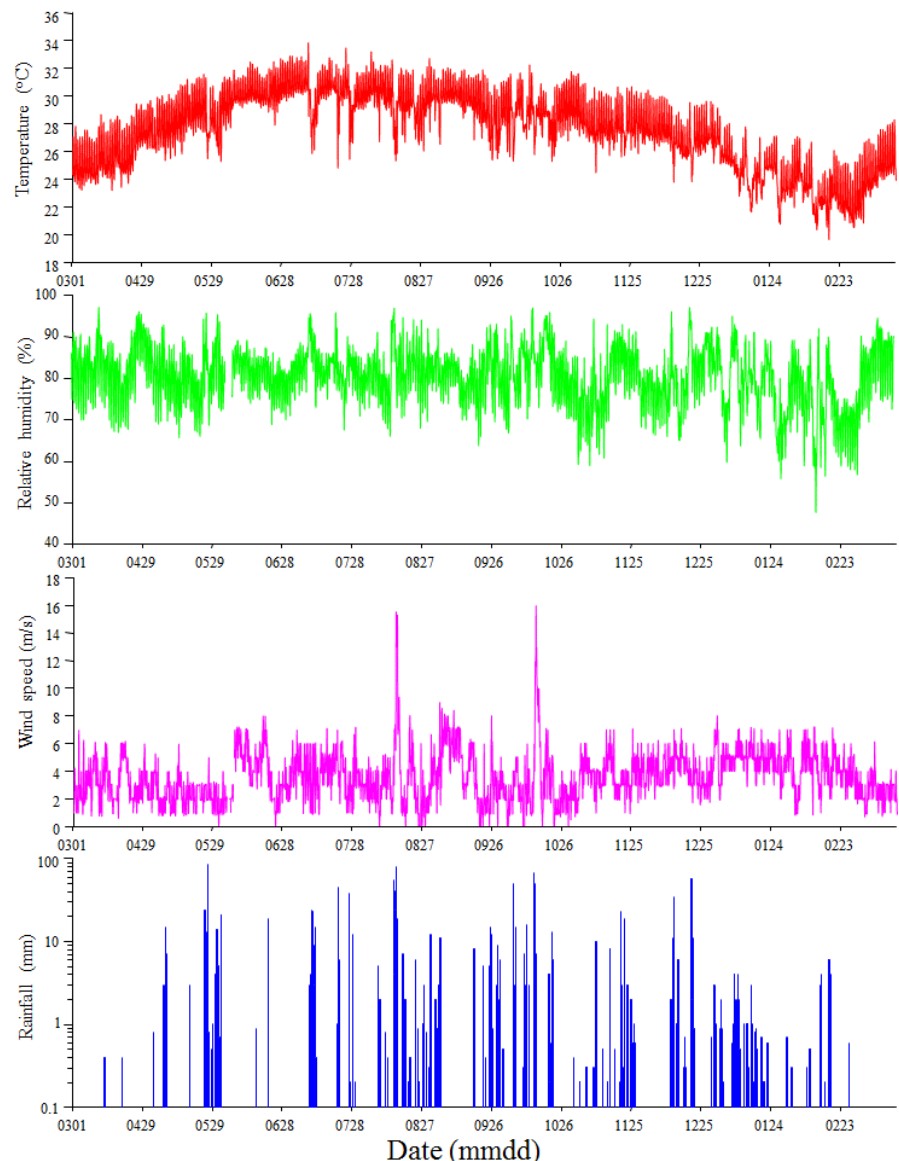



**Figure 3** Annual average mass concentrations of aerosol chemical species at Yongxing Island.

Box boundary indicates 25th and 75th percentile. Lines within the box show the mean. Whiskers

above and below the box indicate 90th and 10th percentiles.

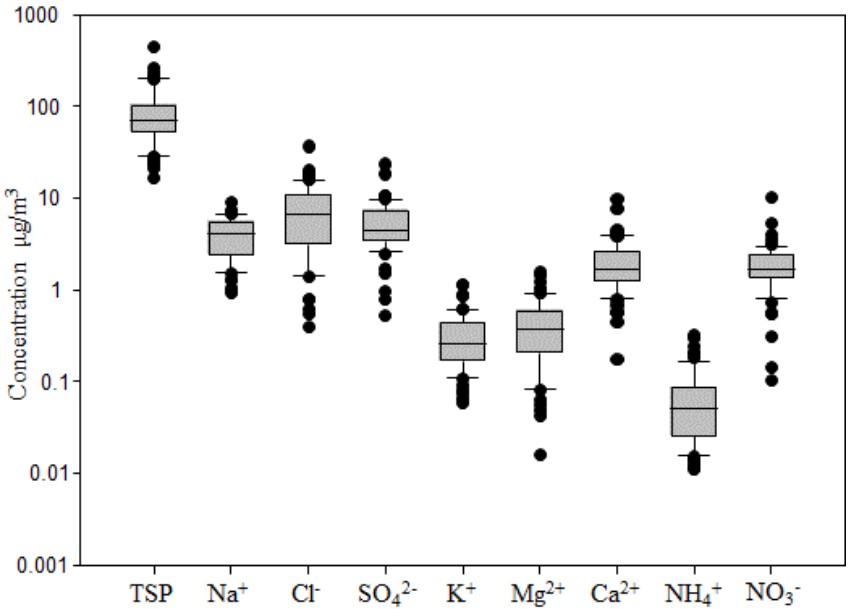





**Figure 4** Mean contributions of each major ionic component to total ionic mass concentration

of (a) Yongxing Island (YXI) annual, (b) YXI cool season, (c) YXI warm season, and (d) YXI

transition season.

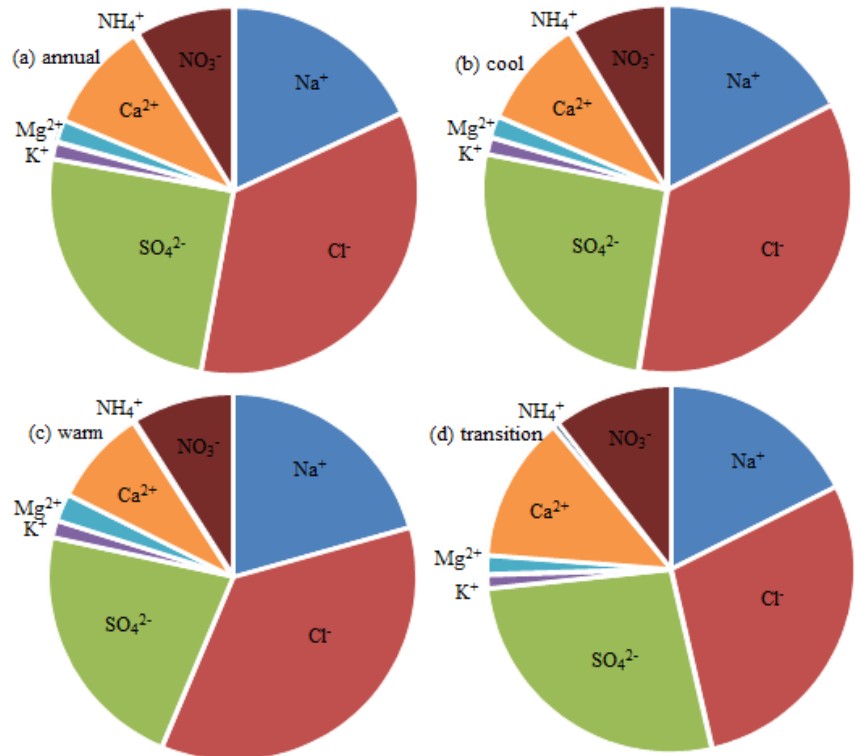





**Figure 5** Comparisons of major ions in aerosol at Yongxing Island with global ocean. Data of Oki, Ogasawara and Hedo are from EANET (www.eanet.asia);

those of Rishiri Island are from Okuda et al. (2006); those of Hawaii are from Carrillo et al. (2002); those of Bermuda are from Moody et al. (2014); those of

Amsterdam Island are from Claeys et al. (2010); those of the Arabian Sea and Indian Ocean are from Kumar et al. (2008); those of Helgoland are from Ebert et

al. (2000); those of the Mediterranean Sea, northern Atlantic-1 and 2, Pacific and southern Atlantic are from Zhang et al. (2010). Pentagrams represent sampling

sites on islands; others represent cruises. N.A. indicates no data.

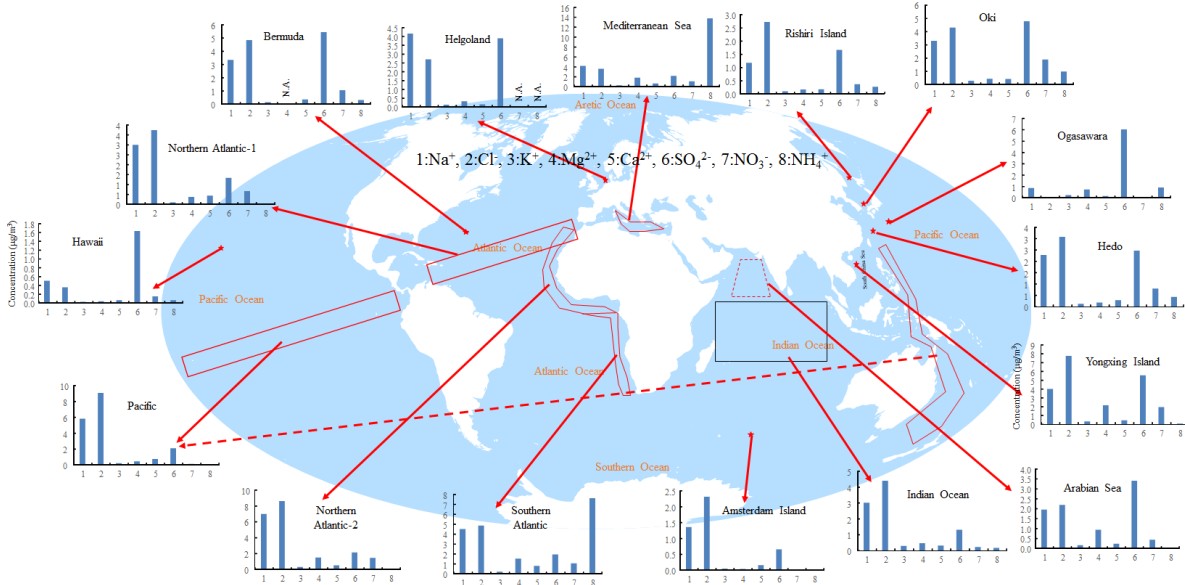



**Figure 6** Seasonal variations of TSP mass concentration and associated species, including Na$^+$, Cl$^-$, SO$_4^{2-}$, Mg$^{2+}$, K$^+$, Ca$^{2+}$, NH$_4^+$, and NO$_3^-$ at Yongxing Island

(cool season: C; warm season: W; annual: A). Shown are the mean and standard deviation for each bar.

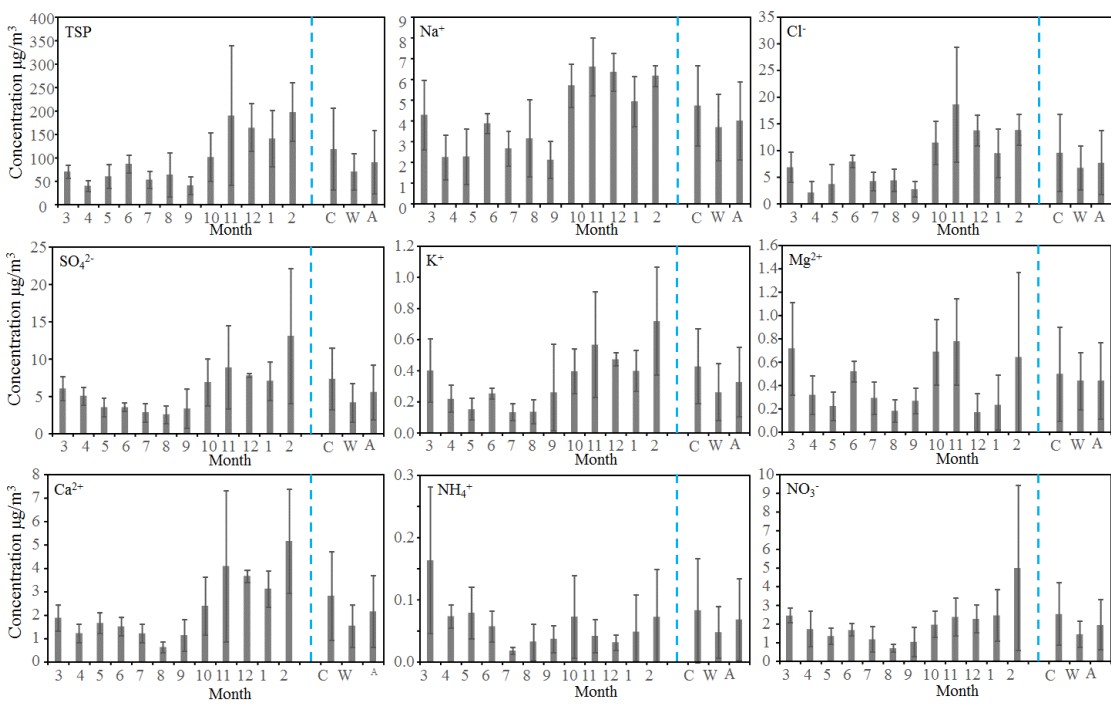



**Figure 7** Comparison of aerosol chemical species between Yongxing Island and around the South China Sea (data from EANET).

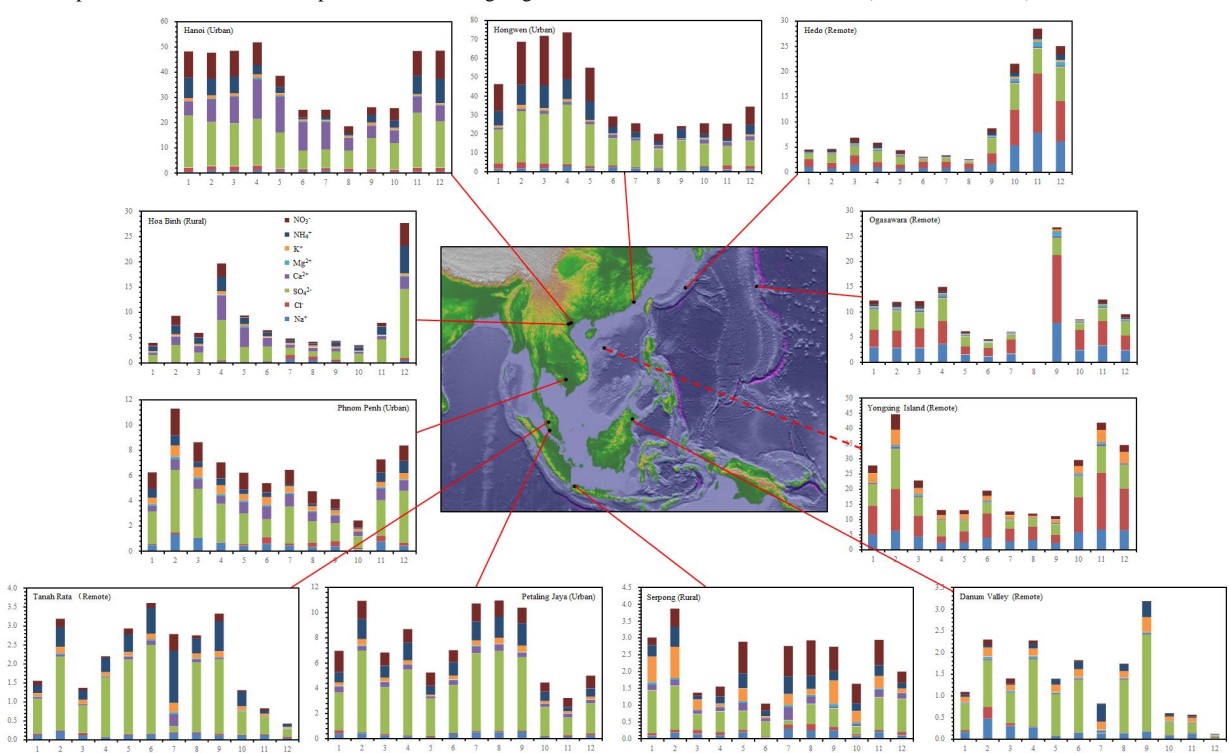





**Figure 8** Fire spot data from MODIS global fire mapping from March 2014 to February 2015 around South China Sea
(https://firms.modaps.eosdis.nasa.gov/firemap/).

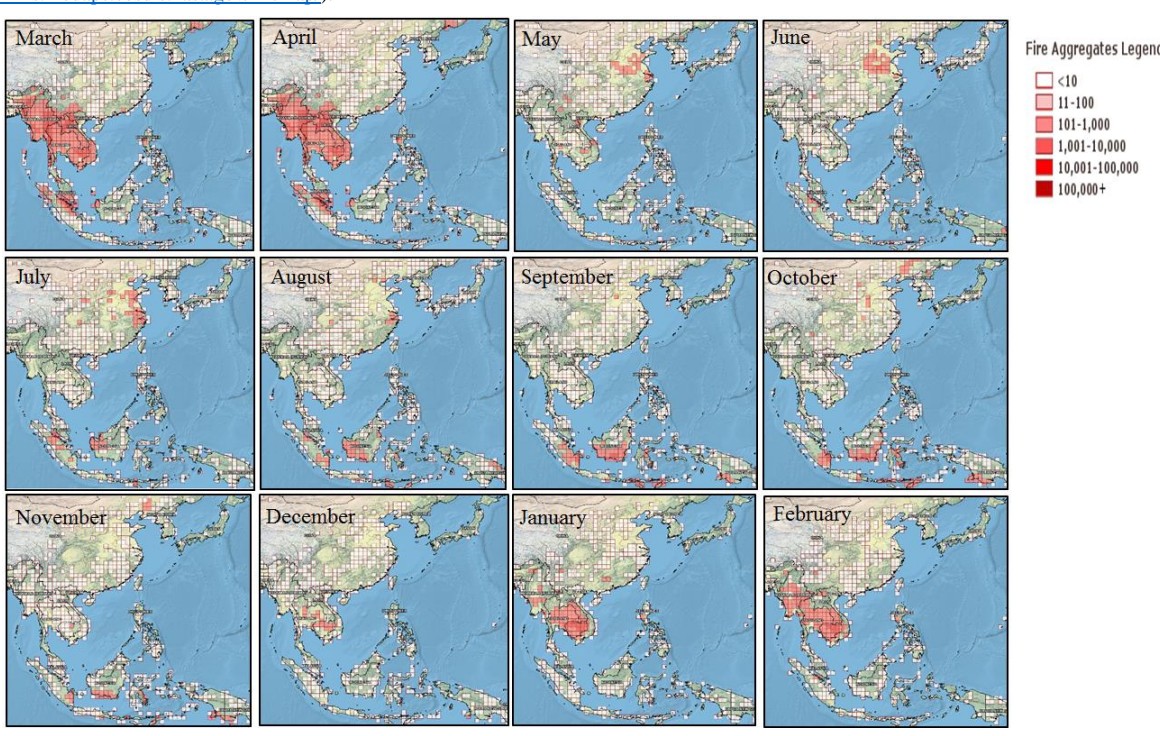




**Figure 9** Plots of (a) principal component analysis (PCA) and (b) classical multidimensional scaling (CMDS) of correlation coefficients among major ions in

aerosol samples.

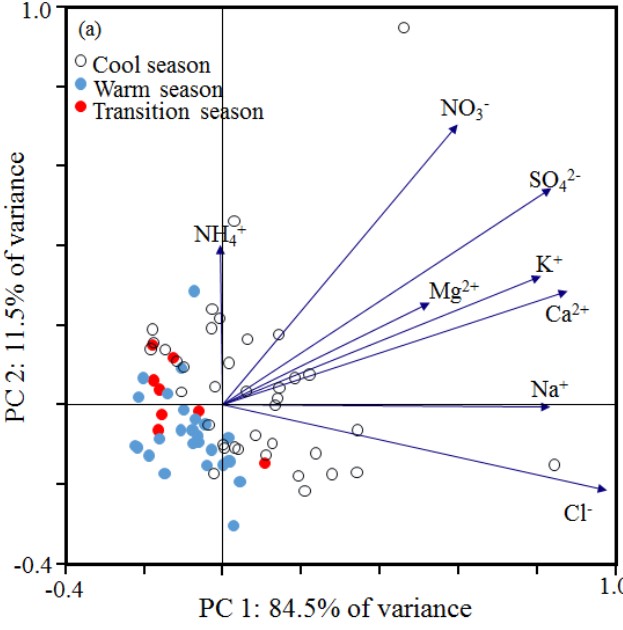
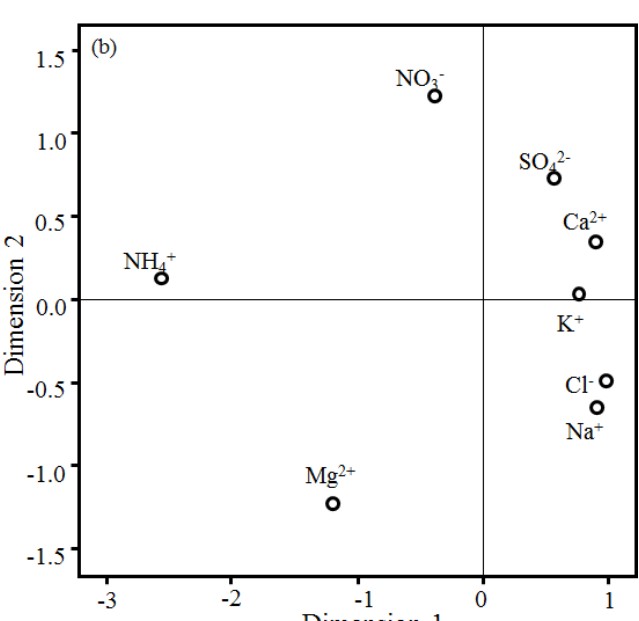

**Figure 10** Ten-day back trajectories of warm (black, June through September 2014), cool (blue, March and April 2014 and October 2014 through February
2015) seasons, and transition season (red, May 2014) at Yongxing Island. Additionally, CWT (concentration weighted trajectory) plots for daily weighted-
average concentrations of TSP, $Ca^{2+}$, $Mg^{2+}$, $K^+$, $SO_4^{2-}$, $NO_3^-$ and $NH_4^+$ at Yongxing Island.

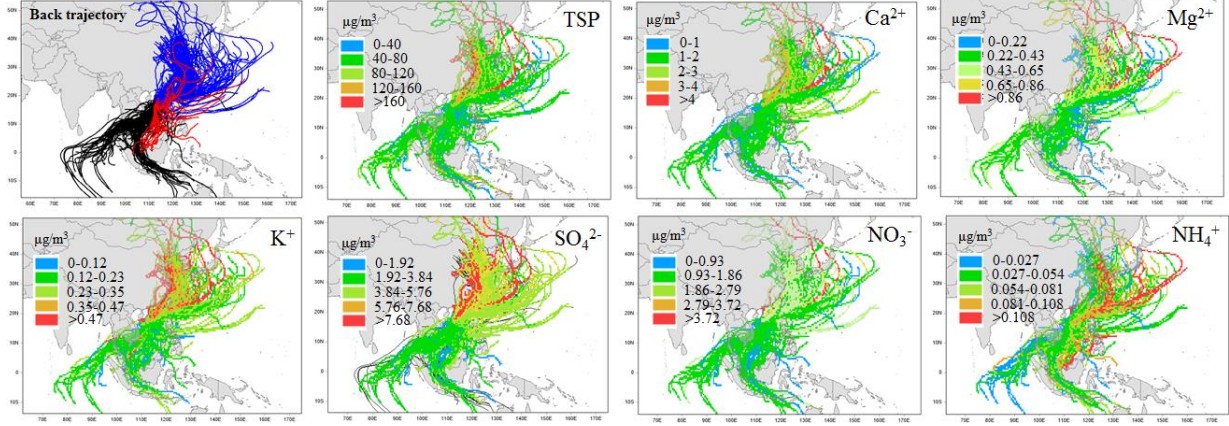
