# Peer review of "Atmospheric aerosol compositions over the South China Sea: Temporal variability and source apportionment"

_Atmospheric Chemistry and Physics, 2016_

## Referee Comment (RC1) · Anonymous Referee #1 · 6 Dec 2016

The manuscript by Xiao et al. presents a very detailed and comprehensive study of total suspended particulates (TSP) in the South China Sea. TSP was collected for the period of one year, covering all seasons, and analyzed for major ions. A variety of source apportionment methods, such as correlation analysis, principal component analysis, back trajectory analysis and positive matrix factorization, were applied to reveal the regional and source-specific origins of TSP. In addition, results are compared to previous studies from the literature and put into wider context.

Generally, this study is of scientific interest as it provides lots of detailed information on TSP in a region where various types of anthropogenic pollution as well as natural emissions from the sea contribute to the local aerosol load. However, this study shows

a lack of methodological detail, the discussion is partly redundant and confusing, and a coherent storyline is missing. While this work is certainly worthwhile to be published, I recommend major revisions as detailed below.

General comments

1. The applied analysis methods must be explained in more detail. In particular, there is no information on the methodologies behind concentration weighted trajectories (CWT), the principal component analysis (PCA) and positive matrix factorization (PMF). It is not sufficient to provide references without explaining the methodology in the text. The reader must be able to understand what the authors did, on a general level, without consulting further literature. In addition, there is no information on how many blanks were produced and in which intervals. In the following I will elaborate a bit on how the PMF related part can be improved. I have less expertise for PCA and CWT but would recommend that the authors check very carefully what the standard for reporting is in the literature and include the respective information this in the manuscript. For instance, with which program were the back trajectories run, Hysplit, Flexpart, Lagranto or other.? What are the uncertainties in relation to the covered distance from the receptor?

2. With regards to PMF, it is well established in the literature which aspects need to be explained at least (e.g., Zhang et al., 2011). In the presented manuscript, the authors do not describe how they prepared the error matrix and especially how they dealt with combining errors from different measurement techniques (i.e. TSP vs major ions). This can be very difficult and has a large effect on the results, please see for example (Crippa et al., 2013) for details. Did the authors downweight any component of the input matrix? In fact, the input matrix is not even described. Furthermore, the authors do not discuss how many solutions they explored (e.g. 1-10 solutions), the number of fpeaks and seeds and their range etc.

With regards to reporting of PMF results, here again a large discrepancy exists between what one would expect to see and what is actually reported (please see again Zhang et al., 2011). For example, as an absolute minimum the time series and profiles of the chosen factors need to be shown and discussed. Based on the presented information, I am unable to review the credibility of the presented results, because in addition to lacking methodological information, I do not know how similar or different the resulting factors are. What are the correlation coefficients between the factor time series and profiles? How do these factors relate to external variables, e.g. meteorological parameters? On the basis of what are the selected factors justified? Etc. All this information needs to be included, before the manuscript can be considered for publication.

3. The manuscript is lengthy. This is in part due to redundancy in the discussion of results from different source apportionment methods, see specific comments. I suggest shortening the discussion section and focusing on a few findings instead of discussing all details. The manuscript is partly confusing for the reader and in the end it is not clear what the main points are. A consistent story line needs to be crafted.

4. The authors use a suite of source apportionment techniques, however it is not clear what the added value is. This is due to the fact that the results are discussed one after another separately per method and no connection between them is established. Often this results in repetitive discussion. Each technique has its strengths and weaknesses that are hardly exploited in this work. When applying so many methods, I would expect that e.g. the CWT are used to supplement PMF results where the PMF results show ambiguities, or that PCA is used in addition to CWT because CWT cannot determine specific source types which PCA can help with. Conversely, CWT are helpful to determine regional provenance of TSP which PCA or PMF cannot provide. Also, in some instances, results are contradictory (see specific comments), this is however not discussed. Such discrepancies need to be addressed rather than focusing only on confirmative results.

5. As indicated in the specific comments sections, references are sometimes missing,

while in other instances it is not clear what exactly the authors refer to in a study when providing a reference.

Specific comments:

l. 30f: It is not clear what you mean by "$Na^+$ and $Cl^-$....made up 74 % and 82%..." These numbers clearly don't add up and information on the reference is missing.

l. 31f: What is marine aerosol in this context? How was it determined?

l. 34: Already in the abstract $NH_4^+$ is claimed to originate from marine biogenic sources. However, throughout the manuscript there is no explanation what these marine biogenic sources are, which seasonality they follow and how the measured ammonium is related to marine biological activity. Without this information, I am not convinced that the ocean is the primary source of ammonium.

l. 38f: what about the role of climate? This first sentence could use some more references since many factors are mentioned.

l. 40: What are "complex sources"? I could imagine that the authors wanted to express that aerosols have many sources which create a complex mixture of aerosol components? What about mineral dust emissions from wind opposed to rock weathering? Also, references are missing.

l. 44: I find the list of aerosol components random. E.g. organics are not mentioned while it has been shown that they constitute an important fraction of aerosol chemical components. Also BC is not mentioned.

l. 45: This statement is not differentiated enough. Some parts of the world have undergone significant socio-economic growth in the past decades, such as East Asia, which has led to much higher emissions. In other parts of the world, emissions have decreased due to stricter air quality legislation. This should be reflected in this sentence or the focus should clearly be on East Asia, the region relevant for the South China Sea (SCS).

[Figure]

l. 50 – 65: The purpose of this paragraph is not clear. What is the point of discussing aerosol deposition and ocean productivity in the context of this particular manuscript? If the idea was to provide a brief review of particulate pollutants to the ocean atmosphere it is not clear why only nitrogen containing compounds are mentioned? Also I do not see the value of reporting observation from many different locations. I would suggest focusing on what is known about the SCS and report on aspects that are of relevance to TSP observations as presented in this manuscript.

l. 75: What is the difference between "aerosols and pollutants" in this context? Do the authors want to distinguish between natural and anthropogenic sources or particulate and gas phase pollutants?

l. 79: Is there not a more recent reference for biomass burning emissions and resulting deposition?

l. 95 f: a reference is missing.

l. 97: "In the present study, it was. . ." What is meant by "it"?

l. 101: What is the "local southeast"?

l. 101: Since the variations of temperature and the difference between what is called the "cold" and "warm" seasons are very small some more information is needed on how seasons were separated and why. Especially what qualifies as transition season?

l. 122f: what do the relative standard deviations refer to? Repeated measurements of a standard, a blank or something else? What about the number of blanks that were generated in the course of the year? Please include more detailed information.

l. 139: In how far do these references reflect what the authors did? These references point towards different tools for running PMF.

l. 149: TSP mass concentrations are compared to those in other cities. The authors write "around the world", however the references point only towards Asian cities. It is

fine to compare to Asian cities only, but this should be made explicit, i.e. state the locations and reference TSP concentrations there.

l. 151f: Again provide numbers for reference. Where are those places, why are they comparable?

l. 155f: This paragraph is not readable. A table is preferable.

l. 162: What does the value in parenthesis represent? An annual average? What is the standard deviation?

l. 164: Again providing numbers for references is needed.

l. 166: Here Fig. 5 is mentioned, while Fig. 4 has not yet been referred to. Please check the order of the figures.

l. 172: What is the "global ocean"? This expression is used various times. Please replace it by a more accurate description of what is meant, e.g. "among all locations"

l. 174: Are the dead corals under water or exposed to the atmosphere? If they are not exposed, I don't understand how they can contribute to the measured calcium.

l. 175: Starting from here, the authors refer to some major ions as non-sea salt ions. However, it is not explained in the manuscript how sea salt and non-sea salt contributions to ions were determined. Please include this information in the methods section.

l. 179: How can the authors show that the Sahara Desert is a source of dust for the measurement location? The way the information is provided is not convincing.

l. 179 f; l. 202: What do the authors want to say with "average Mg 2+ concentrations" being "nearly consistent with...Na+"? Or NO3- concentrations "were often consistent with those of nss-SO4 2-". Do the authors refer to the ion balance? Please explain and change the formulation in the manuscript.

l. 201: references are missing.

l. 216f: "most other studies". Are there only the four that are cited or more? What are their locations? What are those studies about?

l. 222: Why "many other studies" when only two are cited. Again, which locations do these studies refer to?

l. 223: How can TSP and rainfall not be related if some major ions are influenced by TSP?

l. 228: What is meant by "particle wetting and interaction"? From the previous paragraphs I understand that there is more rain during the warm season. So my guess would be that particle activation and scavenging is happening. Do the authors refer to aerosol cloud interactions?

l. 241: This suggests, it doesn't show.

l. 255 before and after: It is not clear to me, why the authors do not discuss the concentrations and ratios of major ions that may originate from sea salt in the context of their ratios in sea salt. The authors even provide a table with typical major ion ratios in sea salt but do not refer to it. The discussion could highly benefit from this addition at this point.

l. 257: What do the authors mean with "complex"?

l. 258: Please specify what is meant with "phenomenon".

l. 259: How does the study of Moody et al. compare with this work? Why is it comparable?

l. 269: Biomass burning is not a major source of sulfur containing species compared to other sources. Why do the authors refer several times to biomass burning as source of SO2 in the manuscript?

l. 276: I suggest reformulating this sentence: "Lawrence and Leliveld (2010) attributed x % of Nox emissions to..." In the current form it sounds like these values were recently

measured.

l. 291: which time period is reflected?

l. 296: What about the influence of anthropogenic activities?

l. 330: What are "dynamic" smoke surface concentrations?

l. 352: "Figure S2 confirms these findings" by showing and proving what?

l. 363: Do the authors mean "accumulation mode" aerosol?

l. 369: After reading this long description I lost track of what the main message is. This needs to be written much more concisely by focusing on the most important findings.

l. 381: "depletion probably occurred". There is no evidence for it.

l. 384f: This conclusion is not evident. How can Cl- from KCl be more dominant than Cl- from sea salt? Furthermore, I do not understand what the difference is to what has been discussed before with regards to K in l. 330-338 (K as marker for biomass burning). This is confusing.

l. 387: I am not convinced that SO4 2- is a biomass burning marker. The relation between potassium and sulfate might result from the transport of air masses from the same source region with different source types.

Section 3.3.1: I suggest integrating the findings from the correlation analysis into the other sections. This section is very redundant and makes the manuscript unnecessarily long.

l. 404: This statement is disconnected from the previous analysis. What has been mentioned that relates to this section?

l. 408: An explanation for what CMDS is, is needed.

Section 3.3.3: Please see comments above. The lack of information and figures is not acceptable.

l. 439: Why 50 % now, in l. 422 it was 58 %.

l. 441: I do not understand this sentence "CaSO4 and sulfate containing both K and Ca..."

l. 449f: Do the authors say that 41 % of potassium comes from biomass burning?

l. 455: "In addition, biomass burning produces SO2 and NOx..." has been mentioned at least for the third time. Again, there are too man repetitions in this manuscript.

l. 460: What is the reason for it, a meteorological situation that favor southward transport of air masses? Again, in many cases more precise information is needed what the authors refer to exactly in the given literature.

l. 485: I do not see the point of "a major discovery". An explanation is needed why the authors think this is new knowledge.

l. 515-520: The origin of ammonium and ammonia is discussed again here. This is repetitive and it is not clear to me, why the authors reveal the information on the nutrient situation in the marine water only at this very late point in the manuscript?

  Technical comments:

l. 19: "major inorganic ion concentrations" instead of "inorganic chemical ionic concentrations"

l. 25: insert "which were" before "higher in the cool season..." and remove the ","

l. 26: finish the sentence after "seasons" and start a new one with "Factors of influence were..."

l. 33: write "was the dominant source of..."

l. 40 f: write "Aerosols have many sources. Primary aerosols, emitted directly from..."

l. 73: remove "SCS" behind "northeast".

l. 94: replace "such" with "the high"

l. 95: replace "be" with "arrive"

l. 103: "Accumulated annual rainfall..."

l. 113: remove "an" before "another"

l. 120, 123, 124: "relative" instead of "relatively"

l. 145, 160: "over the SCS"

l. 147: remove "aerosols" behind "TSP".

l. 165: "annual average TSP and ionic concentration are comparable to..."

l. 172: replace "composed" by "contributes"

l. 205: remove "that"

l. 215: "distinct"

l. 221: I suggest to write: ".. because 70 % of rainfall at...happens during the warm seasons..

l. 224: replace "that" by "mass".

l. 225: replace "strong" by "high"

l. 247: "in contrast to"

l. 254: "correlation" instead of "correction"

l. 264: "suggest" instead of "show".

l. 267: insert "were observed" after (Wang et al. 2006).

l. 276: "emissions" instead of "emission", twice

l. 285: replace "difficult" by "limited"

l. 291: insert "the" before" Acid Deposition. . ."

l. 313: "Excess Cl- in January has been observed by . . ."

l. 316: insert "for almost all stations" at the end of the sentence.

l. 318: no "s" in "oceans"

l. 340: remove "the reported by"

l. 383: insert "fuel" after "fossil"

l. 395: Table 2, I believe. Delete "that".

l. 438: Remove "absolutely"

l. 491: replace "as that" by "compared to"

l. 532: "to help better understand their chemical. . ."

l. 535: "with higher concentrations in the. . ."

  References:

Crippa, M., Canonaco, F., Slowik, J. G., El Haddad, I., DeCarlo, P. F., Mohr, C., Heringa, M. F., Chirico, R., Marchand, N., Temime-Roussel, B., Abidi, E., Poulain, L., Wieden-sohler, A., Baltensperger, U., and Prévôt, A. S. H.: Primary and secondary organic aerosol origin by combined gas-particle phase source apportionment, Atmos. Chem. Phys., 13, 8411-8426, 2013.

Zhang, Q., Jimenez, J. L., Canagaratna, M. R., Ulbrich, I. M., Ng, N. L., Worsnop, D. R., and Sun, Y. L.: Understanding atmospheric organic aerosols via factor analysis of aerosol mass spectrometry: a review, Anal. Bioanal. Chem., 401, 3045-3067, 2011.

---

## Referee Comment (RC2) · Anonymous Referee #2 · 6 Dec 2016

The manuscript presents results from observations of Total Suspended Particulate matter (TSP) from an approximately year long study in a marine region of the northern South China Sea (SCS). Four-day filters were collected and analyzed for major inorganic chemical ionic concentration observed in the marine boundary layer. Several source apportionment methods were then used to both differentiate between aerosol types that contributed to measured values and link them with potential sources. These included correlation between various factors, principal component analysis, positive matrix factorization, and backtrajectory analysis by means of concentration weighted trajectories for various identified sources.

The results are interesting and provide new findings that contribute to knowledge of

aerosol impacts on the northern SCS. However, the manuscript lacks a clear justification of the basis for the study or description of the implications of its findings. In addition, further information on the methods used is needed in order to fully ascertain how the study was conducted and if the findings are justified from the measurements and information described. I therefore recommend that the authors conduct major revisions to the manuscript to focus on providing more reasoning behind why the study was conducted and what conclusions regarding marine aerosol impacts are justified from TSP observations. In addition, the manuscript should be reorganized to more clearly link each part of the methods, results, and conclusions to the overall purpose of the study.

General Comments: 1. The authors utilized only measurements of ionic concentrations of TSP and meteorological information to investigate aerosol sources and impacts on a remote marine region. This provides interesting information, but as filter collections of TSP can be dominated by larger aerosol particles, the authors should briefly discuss the limitations of such measurements in comparison to size resolved measurements.

2. The methods section 2 did not contain enough information to fully describe how the study was conducted. While the cited studies are helpful and needed, at least a basic description of each method, along with the specific parameters of the method used in this study are needed. Specific examples are included in the specific comments.

3. Several sections of the results are repetitive, and some of the sections contain information that would be more helpful to the reader by including it in the methodology section before the results are presented. Streamlining the results section to more clearly describe the results first, followed by a discussion of their implications might allow the reader to follow the logic of the study better. The results section in particular could be better organized in a way such that the results directly lead to descriptions of the findings. Specific examples are included in the specific comments.

4. The authors might consider a more thorough description of the justification for the

study, and specifically why TSP measurements are appropriate for identification of source types impacting remote marine regions of the SCS. The reason for inclusion of some parts of the manuscript was not immediately evident, and did not always lead to a coherent storyline or scientific narrative. A more concise description of what was conducted and why would alleviate many of these concerns.

Specific Comments:

Line 30. It is not clear what "74% and 82%" are referring to in the abstract. Similarly, percentages in the rest of the abstract should be clearly described.

L 43-45. Organics also constitute an important source, and should be mentioned even if measurements were not conducted.

L 70. The description of "warm" and "cool" periods of the monsoon are not clearly linked to the description of the Boreal seasons.

L 100. Consider changing "major" to another term such as "primarily".

L 106; Section 2.2. Additional description on how samples were collected is needed. What height was the inlet? What were the inlet dimensions and type? How was representative sampling of the aerosol assured? Importantly: Were there any local sources at the island that could contribute to TSP and skew results in comparison to background SCS marine boundary layer conditions? How were they identified and/or removed from the results? If other studies have considered this for the location, please note this and cite the study.

L 123 and 124. Should be relative standard deviation?

L 127; Section 2.3. Additional information is needed on the backtrajectory model used, the version, the meteorological dataset source used, the receptor height, and how often backtrajectories were run. A brief description of why 10 day backtrajectories were selected would be helpful as well. Detailed information is not required and can be referred to the cited works, but a brief description would be helpful. Similarly, a brief

description of how CWTs were used would be helpful.

L 136; Section 2.4. Similarly, more detail on the PMF model setup is needed. For instance, a brief description of how and why five sources were selected by the methodology would be helpful for the reader.

L 153. Consider more clearly specifying that percentages are on a mass basis throughout the manuscript. This would assist the reader in more clearly describing what percentages are referring to and how they should be interpreted.

L 160; Section 3.1.2. Results from this section, section 3.1.1., and section 3.2 all contain somewhat repetitive, thought slightly different descriptions, of similar results. Consider reorganizing the results in a way that presents this information, then discusses different relevant findings in a more ordered manner that leads to the study conclusions.

L 223. The lack of correlation between rainfall observed at the receptor and TSP may not be sufficient to warrant the finding that "rainfall is not a major factor controlling seasonal variation of that concentration". Precipitation is a complex process that can lead to both increases in TSP (e.g. via gust fronts etc.) and decreases through wet deposition, among other processes. In addition, relevant processes can occur on time scales below the four-day sample period of the filter samples. A more nuanced discussion of the relationship between precipitation and TSP should be included.

L 226. State the hypothesized mechanism that links decreased TSP to higher temperatures and RHs via particle hygroscopic growth and interactions. Is this finding justified by the available data or are there better source to support this finding? A significant correlation is not sufficient to justify this statement.

L 371; Section 3.3.1. Methods for this section could be included earlier in the methods section 2. It may help to better understand these results when they are first discussed in earlier results sections.

L 414. It can be very difficult to form valid conclusions on the composition of various particle sizes or modes from TSP observations on their own. These statements are somewhat speculative in nature, even with other studies to cite. Additional evidence or discussion should be included to support any contentions based on distinct sources associated with size distribution differences or size segregation.

L 418; Section 3.3.3. More discussion on how PMF results were linked to the identified source types would be helpful. Was this solely based on relative ion concentration? This could be added to the methods section as well.

L 450. Smoke can be an important source of aerosol into the southern SCS during the "warm" monsoonal season as extensive burning can occur in Borneo and Sumatra. That less evidence of this impact is found (due to the noted potassium ratios) in the study's more northern SCS marine region is interesting. Similar impacts from anthropogenic pollution are likewise noteworthy. The authors may wish to spend some time in the discussion emphasizing that there are important sources of aerosol throughout SE Asia and the maritime continent, while the CWT and ionic ratios indicate that sources important to the southern parts of the SCS may be removed or less important to northern SCS regions than those of regions around SE Asia and China.

---

## Author Comment (AC1) · 26 Dec 2016

The manuscript by Xiao et al. presents a very detailed and comprehensive study of total suspended particulates (TSP) in the South China Sea. TSP was collected for the period of one year, covering all seasons, and analyzed for major ions. A variety of source apportionment methods, such as correlation analysis, principal component analysis, back trajectory analysis and positive matrix factorization, were applied to reveal the regional and source-specific origins of TSP. In addition, results are compared to previous studies from the literature and put into wider context.

Generally, this study is of scientific interest as it provides lots of detailed information on TSP in a region where various types of anthropogenic pollution as well as natural emissions from the sea contribute to the local aerosol load. However, this study shows a lack of methodological detail, the discussion is partly redundant and confusing, and a coherent storyline is missing. While this work is certainly worthwhile to be published, I recommend major revisions as detailed below.

**General comments**

1. The applied analysis methods must be explained in more detail. In particular, there is no information on the methodologies behind concentration weighted trajectories (CWT), the principal component analysis (PCA) and positive matrix factorization (PMF). It is not sufficient to provide references without explaining the methodology in the text. The reader must be able to understand what the authors did, on a general level, without consulting further literature. In addition, there is no information on how many blanks were produced and in which intervals. In the following I will elaborate a bit on how the PMF related part can be improved. I have less expertise for PCA and CWT but would recommend that the authors check very carefully what the standard for reporting is in the literature and include the respective information this in the manuscript. For instance, with which program were the back trajectories run, Hysplit, Flexpart, Lagranto or other.? What are the uncertainties in relation to the covered distance from the receptor?

Response: Thank you for your suggestion. We have added more information about the applied analysis methods in Section 2.4 and 2.5 (see detail information in supplementary text S2 and S3).

According to your general comments 3 and 4, we delete the analysis methods of PCA.

2. With regards to PMF, it is well established in the literature which aspects need to be explained at least (e.g., Zhang et al., 2011). In the presented manuscript, the authors do not describe how they prepared the error matrix and especially how they dealt with combining errors from different measurement techniques (i.e. TSP vs major ions). This can be very difficult and has a large effect on the results, please see for example (Crippa et al., 2013) for details. Did the authors downweight any component of the input matrix? In fact, the input matrix is not even described. Furthermore, the authors do not discuss how many solutions they explored (e.g. 1-10 solutions), the number of fpeaks and seeds and their range etc.

With regards to reporting of PMF results, here again a large discrepancy exists between what one would expect to see and what is actually reported (please see again Zhang et al., 2011). For example, as an absolute minimum the time series and profiles of the chosen factors need to be shown and discussed. Based on the presented information, I am unable to review the credibility of the presented results, because in addition to lacking methodological information, I do not know how similar or different the resulting factors are. What are the correlation coefficients between the factor time series and profiles? How do these factors relate to external variables, e.g. meteorological parameters? On the basis of what are the selected factors justified? Etc. All this information needs to be included, before the manuscript can be considered for publication.

Response: Thank you for providing literatures and suggestions. We recalculate the PMF results using PMF5.0 (United States Environmental Protection Agency, EPA) according to the methods of Zhang et al. (2011) and EPA PMF5.0 user guide (Norris et al., 2014), more information was described in Section 2.5 and its supplementary text S3.

In the revised manuscript, all ions were re-calculated without TSP concentrations. The uncertainties by ionic species were provided by the analytical library and we used the uncertainties as error matrix (Norris et al., 2014), and used ionic species and sampling time as input matrix. We downweighted Fe and $F^-$ since they had low signal-to-noise ratios (S/N), and there were no excluded species and samples.

In addition, we added two figures of profiles and time series.

Figure 9 Profiles of five sources identified from the PMF 5.0 model, including sea salt (two species),

Figure 10 Time series contributions from each identified sources, including sea salt (two species), crust, secondary inorganic aerosol, and oceanic emission.

3. The manuscript is lengthy. This is in part due to redundancy in the discussion of results from different source apportionment methods, see specific comments. I suggest shortening the discussion section and focusing on a few findings instead of discussing all details. The manuscript is partly confusing for the reader and in the end it is not clear what the main points are. A consistent story line needs to be crafted.

Response: Yes, the manuscript is lengthy. We have streamlined the manuscript from 19 pages to 15 pages, excluding references.

We have deleted Lines 50-65, section 3.3.1 and 3.3.2, and other sentences in pre-revised manuscript. At the same time, we reorganized the manuscript in some sections and moved some information about the methods of Sample collection and chemical analyses (S1), Back trajectories and CWTs analysis (S2), and PMF model (S3) into supplementary text.

4. The authors use a suite of source apportionment techniques, however it is not clear what the added value is. This is due to the fact that the results are discussed one after another separately per method and no connection between them is established. Often this results in repetitive discussion. Each technique has its strengths and weaknesses that are hardly exploited in this work. When applying so many methods, I would expect that e.g. the CWT are used to supplement PMF results where the PMF results show ambiguities, or that PCA is used in addition to CWT because CWT cannot determine specific source types which PCA can help with. Conversely, CWT are helpful to determine regional provenance of TSP which PCA or PMF cannot provide. Also, in some instances, results are contradictory (see specific comments), this is however not discussed. Such discrepancies need to be addressed rather than focusing only on confirmative results.

Response: Thank you for your suggestion. In the revised manuscript, we combined CWT with PMF to explore source identification, apportionment, and region. It was shown in the new section 3.3.

5. As indicated in the specific comments sections, references are sometimes missing, while in other instances it is not clear what exactly the authors refer to in a study when providing a reference.

Response: Thank you. We have added some references in the manuscript.

**Specific comments:**

l. 30f: It is not clear what you mean by "$Na^+$ and $Cl^-$….made up 74 % and 82%..." These numbers clearly don't add up and information on the reference is missing.

Response: We revised the sentence in Lines 31-32.

l. 31f: What is marine aerosol in this context? How was it determined?

Response: Marine aerosols in the paper refer to those sampled at Yongxing Island over the South China Sea.

l. 34: Already in the abstract $NH_4^+$ is claimed to originate from marine biogenic sources. However, throughout the manuscript there is no explanation what these marine biogenic sources are, which seasonality they follow and how the measured ammonium is related to marine biological activity. Without this information, I am not convinced that the ocean is the primary source of ammonium.

Response: We change "marine biogenic sources" to "oceanic emission" in the manuscript. In addition, according the results of CWTs, air masses with high concentrations of ammonium were from the open ocean, so we suggest that ocean may be the primary source of ammonium.

l. 38f: what about the role of climate? This first sentence could use some more references since many factors are mentioned.

Response: Thank you. We have revised the sentence in Lines 39-42.

l. 40: What are "complex sources"? I could imagine that the authors wanted to express that aerosols have many sources which create a complex mixture of aerosol components? What about mineral dust emissions from wind opposed to rock weathering? Also, references are missing.

Response: Yes. We have revised the sentence in Lines 42-47.

l. 44: I find the list of aerosol components random. E.g. organics are not mentioned while it has been shown that they constitute an important fraction of aerosol chemical components. Also BC is not mentioned.

Response: We have added these information to the revised manuscript (Lines 47-49). Thank you.

l. 45: This statement is not differentiated enough. Some parts of the world have undergone significant socio-economic growth in the past decades, such as East Asia, which has led to much higher emissions. In other parts of the world, emissions have decreased due to stricter air quality legislation. This should be reflected in this sentence or the focus should clearly be on East Asia, the region relevant for the South China Sea (SCS).

Response: Thank you for your suggestion. We have revised in Lines 49-51.

l. 50 – 65: The purpose of this paragraph is not clear. What is the point of discussing aerosol deposition and ocean productivity in the context of this particular manuscript? If the idea was to provide a brief review of particulate pollutants to the ocean atmosphere it is not clear why only nitrogen containing compounds are mentioned? Also I do not see the value of reporting observation from many different locations. I would suggest focusing on what is known about the SCS and report on aspects that are of relevance to TSP observations as presented in this manuscript.

Response: Thank you. We have deleted in this paragraph.

l. 75: What is the difference between "aerosols and pollutants" in this context? Do the authors want to distinguish between natural and anthropogenic sources or particulate and gas phase pollutants?

Response: Yes. Thank you.

l. 79: Is there not a more recent reference for biomass burning emissions and resulting deposition?

Response: We have added more two recent references.

l. 95 f: a reference is missing.

Response: Thank you. We have added them.

l. 97: "In the present study, it was…" What is meant by "it"?

Response: The sampling period. We have revised it.

l. 101: What is the "local southeast"?

Response: Local and short air masses. We have revised it.

l. 101: Since the variations of temperature and the difference between what is called the "cold" and "warm" seasons are very small some more information is needed on how seasons were separated and why. Especially what qualifies as transition season?

Response: In the manuscript, we separated the seasons based on the primarily air masses directions. In generally, the temperature was lower when air masses were primarily from northeast; while the temperature was higher when air masses were primarily from southwest. The air masses of transition season were changed from northeast to southwest when cool season changed to cool season. In some year, the air masses of transition season were changed from southwest to northeast when warm season is changing to cool season. But it was not found in our sampling period.

l. 122f: what do the relative standard deviations refer to? Repeated measurements of a standard, a blank or something else? What about the number of blanks that were generated in the course of the year? Please include more detailed information.

Response: Repeated measurements of a standard. Three blank filters were taken from each package (25 filters).

We have added the detailed information in the revised manuscript (supplementary text S1).

l. 139: In how far do these references reflect what the authors did? These references point towards different tools for running PMF.

Response: We recalculate the PMF results using PMF5.0 (United States Environmental Protection Agency, EPA) according to the methods of Zhang et al. (2011) and EPA PMF5.0 user guide (Norris et al., 2014), more information was described in Section 2.5 and supplementary text S3.

l. 149: TSP mass concentrations are compared to those in other cities. The authors write "around the world", however the references point only towards Asian cities. It is fine to compare to Asian cities only, but this should be made explicit, i.e. state the locations and reference TSP concentrations there.

Response: Thank you. We have deleted this comparison.

l. 151f: Again provide numbers for reference. Where are those places, why are they comparable?

Response: Thank you. We have deleted those places.

l. 155f: This paragraph is not readable. A table is preferable.

Response: Thank you. We have added a table.

**Table 1** Annual average, minimum and maximum mass concentrations ($\mu g/m^3$) of TSP and aerosol chemical species at Yongxing Island.

| | TSP | $Na^+$ | $Cl^-$ | $K^+$ | $Ca^{2+}$ | $Mg^{2+}$ | $SO_4^{2-}$ | $NO_3^-$ | $NH_4^+$ |
|---|---|---|---|---|---|---|---|---|---|
| Annual | 89.6 ± 68.0 | 4.00 ± 1.88 | 7.73 ± 5.99 | 0.33 ± 0.22 | 2.15 ± 1.54 | 0.44 ± 0.33 | 5.54 ± 3.65 | 1.95 ± 1.34 | 0.07 ± 0.07 |
| Minimum | 16.4 | 0.90 | 0.39 | 0.06 | 0.17 | 0.02 | 0.52 | 0.10 | 0.01 |
| Maximum | 440.1 | 8.86 | 36.47 | 1.13 | 9.65 | 1.55 | 23.34 | 10.05 | 0.32 |

l. 162: What does the value in parenthesis represent? An annual average? What is the standard deviation?

Response: An annual average. However, we did not found the standard deviation in their papers.

l. 164: Again providing numbers for references is needed.

Response: OK. The revised sentence was shown in Lines 138-142.

l. 166: Here Fig. 5 is mentioned, while Fig. 4 has not yet been referred to. Please check the order of the figures.

Response: Thank you. We have re-checked the order.

l. 172: What is the "global ocean"? This expression is used various times. Please replace it by a

more accurate description of what is meant, e.g. "among all locations"

Response: Thank you. We revised "global ocean" as "among all locations" in the manuscript.

l. 174: Are the dead corals under water or exposed to the atmosphere? If they are not exposed, I don't understand how they can contribute to the measured calcium.

Response: Thank you. We have deleted this.

l. 175: Starting from here, the authors refer to some major ions as non-sea salt ions. However, it is not explained in the manuscript how sea salt and non-sea salt contributions to ions were determined. Please include this information in the methods section.

Response: Thank you for your suggestion. We have added this information in section 2.3.

l. 179: How can the authors show that the Sahara Desert is a source of dust for the measurement location? The way the information is provided is not convincing.

Response: Sorry to make a misunderstanding about this. We revised this sentence in Lines 158-159.

l. 179 f; l. 202: What do the authors want to say with "average Mg2+ concentrations" being "nearly consistent with…Na+"? Or NO3- concentrations "were often consistent with those of nss-SO42-". Do the authors refer to the ion balance? Please explain and change the formulation in the manuscript.

Response: We have revised them in Lines 159-161 and Lines 187-189.

l. 201: references are missing.

Response: We have added them.

l. 216f: "most other studies". Are there only the four that are cited or more? What are their locations? What are those studies about?

Response: Thank you. We have revised them in Lines 202-204.

l. 222: Why "many other studies" when only two are cited. Again, which locations do these studies refer to?

Response: Thank you. We have revised it in Lines 208-209.

l. 223: How can TSP and rainfall not be related if some major ions are influenced by TSP?

Response: Precipitation is a complex process that can lead to both increases in TSP (e.g. via gust fronts etc.) and decreases through wet deposition. In addition, relevant processes can occur on time scales below the four-day sample period of the filter samples (anonymous referee #2's interactive comment).

So, we have deleted this sentence.

l. 228: What is meant by "particle wetting and interaction"? From the previous paragraphs I understand that there is more rain during the warm season. So my guess would be that particle activation and scavenging is happening. Do the authors refer to aerosol cloud interactions?

Response: Thank you. We have revised them in Lines 212-214.

l. 241: This suggests, it doesn't show.

Response: OK.

l. 255 before and after: It is not clear to me, why the authors do not discuss the concentrations and ratios of major ions that may originate from sea salt in the context of their ratios in sea salt. The authors even provide a table with typical major ion ratios in sea salt but do not refer to it. The discussion could highly benefit from this addition at this point.

Response: Thank you for your suggestion. We have added the discussions of the concentrations and ratios in the manuscript. For examples:

As shown in Fig. 5, Tables 1 and 2, similar trends and strong correlation were observed among $Na^+$, $Cl^-$ and $Mg^{2+}$, and the ratios of $Mg^{2+}$ to $Na^+$ in aerosols were close to that in seawater, suggesting that $Mg^{2+}$ may mainly derive from sea salt rather than continental sources.

l. 257: What do the authors mean with "complex"?

Response: We revised "complex" to "different".

l. 258: Please specify what is meant with "phenomenon".

Response: It mean the same with rainwater at Yongxing Island. We revised it in Lines 234-235.

l. 259: How does the study of Moody et al. compare with this work? Why is it comparable?

Response: We have deleted it.

l. 269: Biomass burning is not a major source of sulfur containing species compared to other sources. Why do the authors refer several times to biomass burning as source of SO2 in the manuscript?

Response: Thank you for your suggestion. We have revised them.

l. 276: I suggest reformulating this sentence: "Lawrence and Leliveld (2010) attributed x % of NOx emissions to…" In the current form it sounds like these values were recently measured.

Response: Thank you for suggestion. We have revised and moved to section 3.3, in Lines 375-376.

l. 291: which time period is reflected?

Response: 2011. We have added it.

l. 296: What about the influence of anthropogenic activities?

Response: Thank you. We have added the information in the manuscript in Lines 265-266..

l. 330: What are "dynamic" smoke surface concentrations?

Response: We deleted "dynamic".

l. 352: "Figure S2 confirms these findings" by showing and proving what?

Response: Thank you. We have revised them in Lines 319-319..

l. 363: Do the authors mean "accumulation mode" aerosol?

Response: Yes.

l. 369: After reading this long description I lost track of what the main message is. This needs to be

written much more concisely by focusing on the most important findings.

Response: Thank you for your suggestion. We have deleted section 3.3.1 and reorganized them to other paragraph.

l. 381: "depletion probably occurred". There is no evidence for it.

Response: Thank you. We have deleted it.

l. 384f: This conclusion is not evident. How can Cl- from KCl be more dominant than Cl- from sea salt? Furthermore, I do not understand what the difference is to what has been discussed before with regards to K in l. 330-338 (K as marker for biomass burning). This is confusing.

Response: Thank you. We have deleted it and reorganized them.

l. 387: I am not convinced that SO42- is a biomass burning marker. The relation between potassium and sulfate might result from the transport of air masses from the same source region with different source types.

Response: Yes. Thank you. We have revised it.

Section 3.3.1: I suggest integrating the findings from the correlation analysis into the other sections. This section is very redundant and makes the manuscript unnecessarily long.

Response: Thank you for your suggestion. We have reorganized them to other sections.

l. 404: This statement is disconnected from the previous analysis. What has been mentioned that relates to this section?

Response: We have deleted section 3.3.2 in pre-revised manuscript.

l. 408: An explanation for what CMDS is, is needed.

Response: We have deleted section 3.3.2 in pre-revised manuscript.

Section 3.3.3: Please see comments above. The lack of information and figures is not acceptable.

Response: Thank you for your suggestion. We have added two figures in this section (Figure 9 and

Figure 10).

l. 439: Why 50 % now, in l. 422 it was 58 %.

Response: We have revised them.

l. 441: I do not understand this sentence "CaSO4 and sulfate containing both K and Ca…"

Response: Thank you. We have deleted it.

l. 449f: Do the authors say that 41 % of potassium comes from biomass burning?

Response: Because we re-calculate PMF model, biomass burning is not a primary source and $K^+$ is existed the form of secondary inorganic aerosol.

l. 455: "In addition, biomass burning produces SO2 and NOx…" has been mentioned at least for the third time. Again, there are too man repetitions in this manuscript.

Response: Yes, we have reorganized them.

l. 460: What is the reason for it, a meteorological situation that favor southward transport of air masses? Again, in many cases more precise information is needed what the authors refer to exactly in the given literature.

Response: In cool season, the wind direct is from northeast. We have revised them (see below figure: wind direct and wind intensity over the northwest Pacific in Jan. 2015)

[Figure]

l. 485: I do not see the point of "a major discovery". An explanation is needed why the authors think this is new knowledge.

Response: We agree. We have deleted "a major discovery".

l. 515-520: The origin of ammonium and ammonia is discussed again here. This is repetitive and it is not clear to me, why the authors reveal the information on the nutrient situation in the marine water only at this very late point in the manuscript?

Response: We reorganize this section with PMF model and delete the discussion about nutrient situation.

**Technical comments:**

l. 19: "major inorganic ion concentrations" instead of "inorganic chemical ionic concentrations"

Response: Accepted.

l. 25: insert "which were" before "higher in the cool season…" and remove the ","

Response: Accepted.

l. 26: finish the sentence after "seasons" and start a new one with "Factors of influence were…"

Response: Accepted.

l. 33: write "was the dominant source of…"

Response: Accepted.

l. 40 f: write "Aerosols have many sources. Primary aerosols, emitted directly from…"

Response: Accepted.

l. 73: remove "SCS" behind "northeast".

Response: Accepted.

l. 94: replace "such" with "the high"

Response: Accepted.

l. 95: replace "be" with "arrive"

Response: Accepted.

l. 103: "Accumulated annual rainfall…"

Response: Accepted.

l. 113: remove "an" before "another"

Response: Accepted.

l. 120, 123, 124: "relative" instead of "relatively"

Response: Accepted.

l. 145, 160: "over the SCS"

Response: Accepted.

l. 147: remove "aerosols" behind "TSP".

Response: Accepted.

l. 165: "annual average TSP and ionic concentration are comparable to…"

Response: Accepted.

l. 172: replace "composed" by "contributes"

Response: Accepted.

l. 205: remove "that"

Response: Accepted.

l. 215: "distinct"

Response: Accepted.

l. 221: I suggest to write: "… because 70 % of rainfall at…happens during the warm seasons..

Response: Accepted.

l. 224: replace "that" by "mass".

Response: Accepted.

l. 225: replace "strong" by "high"

Response: Accepted.

l. 247: "in contrast to"

Response: Accepted.

l. 254: "correlation" instead of "correction"

Response: Accepted.

l. 264: "suggest" instead of "show".

Response: Accepted.

l. 267: insert "were observed" after (Wang et al. 2006).

Response: Accepted.

l. 276: "emissions" instead of "emission", twice

Response: Accepted.

l. 285: replace "difficult" by "limited"

Response: Accepted.

l. 291: insert "the" before" Acid Deposition…"

Response: Accepted.

l. 313: "Excess Cl- in January has been observed by…"

Response: Accepted.

l. 316: insert "for almost all stations" at the end of the sentence.

Response: Accepted.

l. 318: no "s" in "oceans"

Response: Accepted.

l. 340: remove "the reported by"

Response: Accepted.

l. 383: insert "fuel" after "fossil"

Response: Accepted.

l. 395: Table 2, I believe. Delete "that".

Response: Accepted.

l. 438: Remove "absolutely"

Response: Accepted.

l. 491: replace "as that" by "compared to"

Response: Accepted.

l. 532: "to help better understand their chemical…"

Response: Accepted.

l. 535: "with higher concentrations in the…"

Response: Accepted.

---

## Author Comment (AC2) · 26 Dec 2016

The manuscript presents results from observations of Total Suspended Particulate matter (TSP) from an approximately year long study in a marine region of the northern South China Sea (SCS). Four-day filters were collected and analyzed for major inorganic chemical ionic concentration observed in the marine boundary layer. Several source apportionment methods were then used to both differentiate between aerosol types that contributed to measured values and link them with potential sources. These included correlation between various factors, principal component analysis, positive matrix factorization, and backtrajectory analysis by means of concentration weighted trajectories for various identified sources. The results are interesting and provide new findings that contribute to knowledge of aerosol impacts on the northern SCS. However, the manuscript lacks a clear justification of the basis for the study or description of the implications of its findings. In addition, further information on the methods used is needed in order to fully ascertain how the study was conducted and if the findings are justified from the measurements and information described. I therefore recommend that the authors conduct major revisions to the manuscript to focus on providing more reasoning behind why the study was conducted and what conclusions regarding marine aerosol impacts are justified from TSP observations. In addition, the manuscript should be reorganized to more clearly link each part of the methods, results, and conclusions to the overall purpose of the study.

**General Comments:**

1. The authors utilized only measurements of ionic concentrations of TSP and meteorological information to investigate aerosol sources and impacts on a remote marine region. This provides interesting information, but as filter collections of TSP can be dominated by larger aerosol particles, the authors should briefly discuss the limitations of such measurements in comparison to size resolved measurements.

Response: Thank you. The term "total suspended particulate" (TSP) has referred in principle to the mass concentration of all particles considered to be airborne. Samplers for "total aerosol"

first emerges during the time when it is widely thought that it is sufficient to simply draw an aerosol sample through an inlet and to collect the particles on a filter. When sampling is required for health-related purposes, we now know that such sampling needs to be carried out with respect to specific particle size fractions.

2. The methods section 2 did not contain enough information to fully describe how the study was conducted. While the cited studies are helpful and needed, at least a basic description of each method, along with the specific parameters of the method used in this study are needed. Specific examples are included in the specific comments.

Response: Thank you for your suggestion. We have added more information in section 2, and supplementary text S2 and S3.

3. Several sections of the results are repetitive, and some of the sections contain information that would be more helpful to the reader by including it in the methodology section before the results are presented. Streamlining the results section to more clearly describe the results first, followed by a discussion of their implications might allow the reader to follow the logic of the study better. The results section in particular could be better organized in a way such that the results directly lead to descriptions of the findings. Specific examples are included in the specific comments.

Response: Thank you for your suggestion.

We have streamlined the manuscript from 19 pages to 15 pages, excluding references. We have deleted Lines 50-65, section 3.3.1 and 3.3.2, and other sentences in pre-revised manuscript. At the same time, we reorganized the manuscript in some sections and described the results first, followed by a discussion, and moved some information into supplementary text.

4. The authors might consider a more thorough description of the justification for the study, and specifically why TSP measurements are appropriate for identification of source types impacting remote marine regions of the SCS. The reason for inclusion of some parts of the manuscript was not immediately evident, and did not always lead to a coherent storyline or scientific narrative. A more concise description of what was conducted and why would alleviate

many of these concerns.

Response: Thank you for your suggestion. We have revised them in manuscript. Such as Lines 19-21 and Lines 65-69:

In order to evaluate impacts of different source emission on marine atmospheric particles over the South China Sea (SCS), major inorganic ionic concentrations ($Na^+$, $Cl^-$, $SO_4^{2-}$, $Ca^{2+}$, $Mg^{2+}$, $K^+$, $NH_4^+$, $NO_3^-$) were determined in total suspended particulates (TSP) at Yongxing Island from March 2014 to February 2015.

However, the observational data on aerosol chemistry over the SCS are very sparse (Xiao et al., 2015).To get better understanding of potential sources, source contributions, and spatio-temporal variations of marine aerosols over the SCS, total suspended particulate (TSP) were continuously collected at Yongxing Island from March 2014 to February 2015.

**Specific Comments:**

Line 30. It is not clear what "74% and 82%" are referring to in the abstract. Similarly, percentages in the rest of the abstract should be clearly described.

Response: Because we re-calculate the PMF model, some values have been changed. We revised the sentence in Lines 31-32.

L 43-45. Organics also constitute an important source, and should be mentioned even if measurements were not conducted.

Response: Yes. Thank you. We have added these information to the revised manuscript in Lines 47-49.

L 70. The description of "warm" and "cool" periods of the monsoon are not clearly linked to the description of the Boreal seasons.

Response: Yes.

In the manuscript, we separate the seasons based on the primarily air masses directions. In generally, the temperature was lower when air masses were primarily from northeast; while the temperature was higher when air masses were primarily from southwest. The air masses of transition season were changed from northeast to southwest when cool season changed to cool season. In some year,

the air masses of transition season were changed from southwest to northeast when warm season is changing to cool season. But it was not found in our sampling period.

L 100. Consider changing "major" to another term such as "primarily".

Response: Thank you. We have revised them.

L 106; Section 2.2. Additional description on how samples were collected is needed. What height was the inlet? What were the inlet dimensions and type? How was representative sampling of the aerosol assured? Importantly: Were there any local sources at the island that could contribute to TSP and skew results in comparison to background SCS marine boundary layer conditions? How were they identified and/or removed from the results? If other studies have considered this for the location, please note this and cite the study.

Response: Thank you for your suggestion. We have added the description in the manuscript (supplementary text S1). We think the samples were representative, because the sampling station is far away from continent at the island with few population and no industry.

L 123 and 124. Should be relative standard deviation?

Response: Yes and thank you.

L 127; Section 2.3. Additional information is needed on the backtrajectory model used, the version, the meteorological dataset source used, the receptor height, and how often backtrajectories were run. A brief description of why 10 day backtrajectories were selected would be helpful as well. Detailed information is not required and can be referred to the cited works, but a brief description would be helpful. Similarly, a brief description of how CWTs were used would be helpful.

Response: Thank you for your suggestion. We have revised this section 2.4 and supplementary text S2.

L 136; Section 2.4. Similarly, more detail on the PMF model setup is needed. For instance, a brief description of how and why five sources were selected by the methodology would be

helpful for the reader.

Response: Thank you. We also have revised this section 2.5 and supplementary text S3.

L 153. Consider more clearly specifying that percentages are on a mass basis throughout the manuscript. This would assist the reader in more clearly describing what percentages are referring to and how they should be interpreted.

Response: Thank you for your suggestion. We have added the information into the manuscript in Lines 128-129.

L 160; Section 3.1.2. Results from this section, section 3.1.1., and section 3.2 all contain somewhat repetitive, thought slightly different descriptions, of similar results. Consider reorganizing the results in a way that presents this information, then discusses different relevant findings in a more ordered manner that leads to the study conclusions.

Response: Thank you. We have revised the sections 3.1.1.

L 223. The lack of correlation between rainfall observed at the receptor and TSP may not be sufficient to warrant the finding that "rainfall is not a major factor controlling seasonal variation of that concentration". Precipitation is a complex process that can lead to both increases in TSP (e.g. via gust fronts etc.) and decreases through wet deposition, among other processes. In addition, relevant processes can occur on time scales below the four-day sample period of the filter samples. A more nuanced discussion of the relationship between precipitation and TSP should be included.

Response: We agree and have deleted this sentence.

L 226. State the hypothesized mechanism that links decreased TSP to higher temperatures and RHs via particle hygroscopic growth and interactions. Is this finding justified by the available data or are there better source to support this finding? A significant correlation is not sufficient to justify this statement.

Response: Thank you for your suggestion. We have added a reference in the manuscript.

We discovered negative correlations between TSP concentration and temperature ($p < 0.01$) and

relatively humidity ($p < 0.01$) (Table 2), indicating that warm temperatures and high relatively humidity enhance particle activation and scavenging is happening (Liu et al., 2011).

L 371; Section 3.3.1. Methods for this section could be included earlier in the methods section 2. It may help to better understand these results when they are first discussed in earlier results sections.

Response: We agree and have deleted this section. But we have reorganized them into other sections.

L 414. It can be very difficult to form valid conclusions on the composition of various particle sizes or modes from TSP observations on their own. These statements are somewhat speculative in nature, even with other studies to cite. Additional evidence or discussion should be included to support any contentions based on distinct sources associated with size distribution differences or size segregation.

Response: Thank you for your suggestion. We have deleted section 3.3.2 "Principal component analysis and classical multidimensional scaling".

L 418; Section 3.3.3. More discussion on how PMF results were linked to the identified source types would be helpful. Was this solely based on relative ion concentration? This could be added to the methods section as well.

Response: Thank you for your suggestion. We have added more information to the methods section.

L 450. Smoke can be an important source of aerosol into the southern SCS during the "warm" monsoonal season as extensive burning can occur in Borneo and Sumatra. That less evidence of this impact is found (due to the noted potassium ratios) in the study's more northern SCS marine region is interesting. Similar impacts from anthropogenic pollution are likewise noteworthy. The authors may wish to spend some time in the discussion emphasizing that there are important sources of aerosol throughout SE Asia and the maritime continent, while the CWT and ionic ratios indicate that sources important to the southern parts of the SCS may be removed or less important to northern SCS regions than those of regions around SE Asia and China.

Response: Thank you. We have re-calculated the PMF model and combined the PMF results with

CWTs results to discuss. In the revised manuscript, we found that K from biomass burning would loss its information when it transport to open ocean, because it can react with $H_2SO_4$ or $HNO_3$ in the atmosphere to form secondary aerosols.

---

## Author Comment (AC3) · 28 Dec 2016

**Atmospheric aerosol compositions over the South China Sea: Temporal variability and source apportionment**

Hong-Wei Xiao[1,2], Hua-Yun Xiao[1,2*], Li Luo[1,2*], Chun-Yan Shen[3], Ai-Min Long[4], Lin Chen[5], Zhen-Hua Long[5], Da-Ning Li[5]

[1]Laboratory of Atmospheric Environment, Key Laboratory of Nuclear Resources and Environment (Ministry of Education), East China University of Technology, Nanchang 330013, China

[2]School of Water Resources and Environmental Engineering, East China University of Technology, Nanchang 330013, China

[3]College of Fisheries, Guangdong Ocean University, Zhanjiang 524088, China

[4]State Key Laboratory of Tropical Oceanography, South China Sea Institute of Oceanology, Chinese Academy of Sciences, Guangzhou 510301, China

[5]Xisha Deep Sea Marine Environment Observation and Research Station, South China Sea Institute of Oceanology, Chinese Academy of Sciences, Sansha 573199, China

*Correspondence to: Hua-Yun Xiao (xiaohuayun@ecit.cn) and Li Luo (luoli@ecit.cn)

**Abstract.** In order to evaluate impacts of different source emission on marine atmospheric particles over the South China Sea (SCS), major inorganic ionic concentrations ($Na^+$, $Cl^-$, $SO_4^{2-}$, $Ca^{2+}$, $Mg^{2+}$, $K^+$, $NH_4^+$, $NO_3^-$) were determined in total suspended particulates (TSP) at Yongxing Island, from March 2014 to February 2015. The annual average concentration of TSP was $89.6 \pm 68.0\ \mu g/m^3$, with $114.7 \pm 82.1$, $60.4 \pm 27.0$, and $59.5 \pm 25.6\ \mu g/m^3$ in cool, warm, and transition seasons, respectively. $Cl^-$ had the highest concentration, with an annual average of $7.73 \pm 5.99\ \mu g/m^3$, followed by $SO_4^{2-}$ ($5.54 \pm 3.65\ \mu g/m^3$), $Na^+$ ($4.00 \pm 1.88\ \mu g/m^3$), $Ca^{2+}$ ($2.15 \pm 1.54\ \mu g/m^3$), $NO_3^-$ ($1.95 \pm 1.34\ \mu g/m^3$), $Mg^{2+}$ ($0.44 \pm 0.33\ \mu g/m^3$), $K^+$ ($0.33 \pm 0.22\ \mu g/m^3$), and $NH_4^+$ ($0.07 \pm 0.07\ \mu 
[revised manuscript text omitted]

*Corresponding author. E-mail: Xiaohuayun@ecit.cn

This PDF file includes:

Supplementary Text: S1, S2 and S3

Reference List

**S1 Sample collection and chemical analyses**

Aerosol was collected on quartz filters ($8 \times 10$ inch, Tissuquartz™ Filters, 2500 QAT-UP, Pallflex, Washington, USA) using a special high-flow rate ($1.05 \pm 0.03$ m$^3$/min) KC-1000

sampler (Laoshan Institute for Electronic Equipment, Qingdao, China), which were installed 1m above the building (about 15m) roof's surface of the station of SCSIO, CAS, and there was not obvious pollution around this station. The sampling time was nominally 96 hours (4 days one sample). Three blank filters were taken from each package (25 filters). All samples and blank filters were stored in a refrigerator at $-20°C$ until analysis in the laboratory.

In the laboratory, one eighth filters were cut and placed in a clean 50-ml Nalgene tube with additional 35-ml ultrapure water. These tubes were washed for 30 minutes using ultrasonic vibration. They were then shaken for 30 minutes on a horizontal shaker at a rate of ~300 rpm and left to rest for another 30 minutes at room temperature. The extract was filtered using pinhole filters, which were then rinsed twice with 5-ml ultrapure water. The extract and rinse were put into 50-ml tubes together and stored in a refrigerator at $-20 °C$ until chemical analyses.

Major anion concentrations ($F^-$, $Cl^-$, $NO_3^-$, $SO_4^{2-}$, $Br^-$) were determined by ICS-90 ion chromatography (Dionex, California, USA). Water-soluble metal and nonmetal elemental concentrations (Al, Ca, Fe, K, Mg, Mn, Na, $SiO_2$, Sr) were analyzed by an MPX inductively coupled plasma optical emission spectrometer (ICP-OES, Vista, CA, USA). $NH_4^+$

concentration was determined by spectrophotometry after treatment with Nessler's reagent. The detection limits of $F^-$, $Cl^-$, $NO_3^-$, $SO_4^{2-}$, $Br^-$ were 0.03, 0.03, 0.08, 0.075 and 0.1 mg/L, respectively, and the relative standard deviation of these ions of standard samples were 0.57%,

2.55%, 1.16%, 1.36% and 11.36%, respectively (Xiao et al., 2013 and 2016). The detection limits of Al, Ca, Fe, K, Mg, Mn, Na, $SiO_2$, Sr were 0.025, 0.003, 0.002, 0.06, 0.0005, 0.0005,

0.02, 0.015 and 0.00008 mg/L, respectively, and the relative standard deviation of these ions of standard samples were less than 1.5% (Xiao et al., 2013 and 2016). The detection limit of $NH_4^+$

was 0.1 mg/L and its relative standard deviation was less than 5.0% (Xiao et al., 2013 and 2016).

In this study, Al and $Br^-$ in most of samples was less than the detection limit.

**S2 Back trajectories and concentration weighted trajectories analysis**

        Back trajectories and concentration weighted trajectories (CWT) are used to determine the long-distance transport of atmospheric pollutants and regional source areas (Cheng et al., 2013;

Xiao et al., 2014 and 2015). The CWT is a good model to estimate potential sources areas, when grid cells are more than 2 trajectory segment endpoint (Cheng et al., 2013). For each day,

10-day (240 hours) back trajectories of air masses (Pavuluri et al., 2015) arriving at Yongxing

Island were computed by the program of TrajStat (version 1.2.26) (Wang et al., 2009). 10-day back trajectories are used in this study since trajectories of a short duration are not long enough to indicate possible distant sources regions (Harris and Kahl, 1990). We also used the program of TrajStat to model CWT of TSP, and $Ca^{2+}$, $Mg^{2+}$, $K^+$, $SO_4^{2-}$, $NO_3^-$ and $NH_4^+$ concentrations at the island. In CWT model, each grid cell receives a weighted concentration obtained by averaging sample concentrations that have associated trajectories crossing the grid cell as follows (Eq. S1; Wang et al., 2009; Xiao et al., 2014 and 2015).

$$C_{ij} = \frac{1}{\sum\limits_{l=1}^{M} \tau_{ijl}} \sum_{l=1}^{M} C_l \tau_{ijl} \qquad \text{(S1)}$$

where $C_{ij}$ is the average weighted concentration in grid cell ($i$, $j$); $l$ is the index of the trajectory;

$C_l$ is the measured ionic concentration, corresponding with the arrival of back-trajectory $l$ at the sampling site; $\tau_{ijl}$ is the time spent in the grid cell ($i$, $j$) by trajectory $l$; M is the total number of back trajectories. The region from 70°E to 160°E and from 20°S to 60°N was defined as the source domain based on back trajectories during the sampling period, containing 14,400 grid cells of $0.5° \times 0.5°$.

**S3 Positive matrix factorization model**

Receptor models are used to quantify the contributions of sources to samples based on the composition or fingerprints of the sources (Norris et al., 2014). The positive matrix factorization (PMF) is an effective source apportionment receptor model that does not require source profiles prior to analysis and has no limitation on source numbers (Crippa et al., 2013; Tiwari et al., 2013; Zhang et al., 2011; Zhang et al., 2015). The PMF model describes the observation ($x_{ij}$) as a linear combination of a number of factors $p$ for each time step $i$ and $j$, whose contribution over time is always positive ($g_k$) and whose mass spectra ($f_k$) are static (see Eq. S2; Crippa et al., 2013; Paatero et al., 2014).

$$x_{ij} = \sum_{k=1}^{p} g_{ik} f_{ki} + e_{ij} \qquad (S2)$$

where $e_{ij}$ is the residual. PMF decomposes the matrix of speciated sample data into two matrices: factor contributions (G) and factor profiles (F) (Norris et al., 2014). G and F are derived by the PMF model minimizing the objective function Q (Eq. S3):

$$Q = \sum_{i=1}^{n} \sum_{j=1}^{m} \left[ \frac{e_{ij}}{u_{ij}} \right]^2 \qquad (S3)$$

where $u_{ij}$ is the measurement uncertainty. In our study, PMF 5.0 (United States Environmental Protection Agency) was used to determine source apportionment of each major ion based on $F^-$, $Cl^-$, $NO_3^-$, $SO_4^{2-}$, $NH_4^+$, Ca, K, Mg, Mn, Fe, Na, and Sr by sampling time, with uncertainties by species provided by the analytical library. However, we downweighted Fe and $F^-$ since they had low signal-to-noise ratios (S/N), and there were no excluded species and samples. We run PMF

for the range of number factors from 2 to 10, and examine the Q(Robust)/Qexp to choose the best model number of factors ($P$). Five physically realistic sources major ions were identified, i.e., sea salt (two species, Na and Mg), secondary inorganic aerosol (SIA; $F^-$, $SO_4^{2-}$ and $NO_3^-$), oceanic emission ($NH_4^+$), and crust (Mn). The model results show that there are good relationships between observed (input data) values and predicted (modeled) values of each major ion except Sr and $F^-$ (correlation coefficients $R^2$ were 0.94, 0.93, 0.94, 0.80, 1.00, 0.93,

0.95, 0.91, 0.59, 1.00, 0.40 and 0.40 for Na, $Cl^-$, $SO_4^{2-}$, K, Mg, Ca, $NH_4^+$, $NO_3^-$, Fe, Mn, Sr, and $F^-$,respectively), and the modeled values of time series are fitting the observed values well. For the best solution chosen above, we run PMF for Fpeak in Rotational Tools (-1.0, -0.8, -0.6, -0.4, -0.2, -0.01, 0.01, 0.2, 0.4, 0.6, 0.8 and 1.0). The results show that the best solution is the

Fpeak near to zero (-0.01 and 0.01) for *P* 5.

---

## Referee Report (RR1)

The authors present a reorganized manuscript of a study of TSP aerosol observations at a location in the northern South China Sea. The focus of the reorganization of the paper has been on presenting a clearer description of the results and a more thorough description of the methodology used in the study. In particular, they have added additional methodological detail in two supplemental sections that provide additional information on sample collection and analysis, and the use of back-trajectory analyses. They have also removed redundant information and presented the results in more clearly defined sections.

The authors adequately addressed most of the earlier concerns raised about the manuscript, though several minor issues remain. Overall, the presentation of the key phenomena in TSP aerosol variability that were observed are now presented effectively. There is not substantial treatment of the limitations TSP aerosol observations have in regards to interpreting the relative importance of various sources in the region. However, this facet of the analysis may not be necessary in order to present information on TSP and its variability in the region, or to comparisons of the results to other regions. I therefore recommend the manuscript to be published subject to minor corrections.

Specific comments:

1. Supplement S1. The additional description in the supplemental information adequately explains the sample collection process. I am used to seeing more of this in the main manuscript methods, but the authors may leave this here as long as they explain this in section 2.2.

2. Supplement S2. I believe TrajStat is an analysis package that is built on top of the NOAA HYSPLIT model. The details of how this was run are still needed. In particular, HYSPLIT version number, the meteorological dataset used (e.g. Gdas1 is often used), and the receptor height. If any sensitivity testing (e.g. time of day) was conducted to validate representativeness of the data, that could be included as well, though for this island receptor it may not be necessary. At minimum, a short mention and reference to HYSPLIT should be included in section 2.4.

3. Use of the NAAPS model for figure S1 should be mentioned in the methods somewhere.

Typos and minor correction suggestions:

L28: "Air mass source region" rather than "air masses"

L32: Should be "77.4% of Na+ and 99.3% of Cl-"

L43: "… anthropogenic sources, such as terrestrial…" could be "… anthropogenic sources, include terrestrial…"

L59: Should refer to boreal winter, northern hemisphere winter, or similar term, rather than just winter.

L61: Again, clarify the "cool season" refers to the boreal winter cool season, or the cool northeastern monsoon phase, etc.

L180: Is "local" intended here? Source regions were from remote ocean regions, rather than local regions?

L187: Should be "important"

L334: Secondary inorganic aerosol for the SIA acronym is not defined earlier.

L429: Consider phrasing as "Chemical composition of one year of aerosol samples…" or similar. Also, the beginning of the conclusion should make clear that results are for TSP aerosol samples (as opposed to PM10, PM1, etc…).

L438: The final conclusion should refer to only the SCS or northern SCS, rather than the wider northern Pacific ocean.

Figure 7. The labels on the chemical species are a bit small and hard to read.

---

## Author Response (AR2)

**Response to anonymous referee #2's comment on the manuscript point by point below.**

The authors present a reorganized manuscript of a study of TSP aerosol observations at a location in the northern South China Sea. The focus of the reorganization of the paper has been on presenting a clearer description of the results and a more thorough description of the methodology used in the study. In particular, they have added additional methodological detail in two supplemental sections that provide additional information on sample collection and analysis, and the use of back-trajectory analyses. They have also removed redundant information and presented the results in more clearly defined sections.

The authors adequately addressed most of the earlier concerns raised about the manuscript, though several minor issues remain. Overall, the presentation of the key phenomena in TSP aerosol variability that were observed are now presented effectively. There is not substantial treatment of the limitations TSP aerosol observations have in regards to interpreting the relative importance of various sources in the region. However, this facet of the analysis may not be necessary in order to present information on TSP and its variability in the region, or to comparisons of the results to other regions. I therefore recommend the manuscript to be published subject to minor corrections.

Specific comments:

1. Supplement S1. The additional description in the supplemental information adequately explains the sample collection process. I am used to seeing more of this in the main manuscript methods, but the authors may leave this here as long as they explain this in section 2.2.

Response: Thank you for your suggestion. We have moved the information of the sample collection process to the main manuscript in section 2.2.

2. Supplement S2. I believe TrajStat is an analysis package that is built on top of the NOAA HYSPLIT model. The details of how this was run are still needed. In particular, HYSPLIT version number, the meteorological dataset used (e.g. Gdas1 is often used), and the receptor height. If any sensitivity testing (e.g. time of day) was conducted to validate representativeness of the data, that could be included as well, though for this island receptor it may not be necessary. At minimum, a short mention and reference to HYSPLIT should be included in section 2.4.

Response: Thank you. We have added more information in Supplement S2 and section 2.4.

Supplement S2: For each day, 10-day (240 hours) back trajectories of air masses (Pavuluri et al., 2015) arriving at Yongxing Island were computed by the program of TrajStat (version 1.2.26) (Wang et al., 2009) with Global Data Assimilation System (GDAS) data (http://www.arl.noaa.gov/HYSPLIT.php) of GDAS one-degree archive (Xiao et al., 2015). The top of model was set to 1000m above sea level.

Section 2.4: For each day, 10-day (240 hours) back trajectories of air masses (Pavuluri et al., 2015) arriving at Yongxing Island were computed, with top of model set to 1000m above sea level (Xiao et al., 2015). In CWT model, each grid cell receives a weighted concentration obtained by averaging sample concentrations that have associated trajectories crossing the grid cell (Xiao et al., 2015).

3. Use of the NAAPS model for figure S1 should be mentioned in the methods somewhere.
Response: We have added the information in section 2.1.
In order to identify the effect of sulfate, dust and smoke on Yongxing Island, we obtain NAAPS Global Aerosol Model data (gif format, https://www.nrlmry.navy.mil/aerosol/#aerosolobservations) between Marth 2014 and February 2015 to synthesize one year of picture (gif, Fig. S1).

Typos and minor correction suggestions:
L28: "Air mass source region" rather than "air masses"
Response: OK.

L32: Should be "77.4% of Na+ and 99.3% of Cl-"
Response: Yes.

L43: "… anthropogenic sources, such as terrestrial…" could be "… anthropogenic sources, include terrestrial…"
Response: Thank you.

L59: Should refer to boreal winter, northern hemisphere winter, or similar term, rather than just winter.
Response: Yes. It refers to "boreal winter", and we have revised them.

L61: Again, clarify the "cool season" refers to the boreal winter cool season, or the cool northeastern monsoon phase, etc.
Response: OK. We used "boreal winter cool season" in the manuscript.

L180: Is "local" intended here? Source regions were from remote ocean regions, rather than local regions?
Response: Thank you. We have revised it.

L187: Should be "important"
Response: Yes.

L334: Secondary inorganic aerosol for the SIA acronym is not defined earlier.
Response: Thank you. We have added "Secondary inorganic aerosol" before "SIA".

L429: Consider phrasing as "Chemical composition of one year of aerosol samples…" or similar. Also, the beginning of the conclusion should make clear that results are for TSP aerosol samples (as opposed to PM10, PM1, etc…).
Response: Thank you. We have revised it as "Chemical compositions of one year of aerosol samples (TSP) at Yongxing Island".

L438: The final conclusion should refer to only the SCS or northern SCS, rather than the wider northern Pacific ocean.
Response: Agree.

Figure 7. The labels on the chemical species are a bit small and hard to read.

Response: Thank you. We have revised them.